# Primary care screening for sexually transmitted infections in the United States from 2019 to 2021

Shiying Hao[1*], Esther E. Velásquez[1], William S. Pearson[2], Karen W. Hoover[3], Weiming Zhu[3], Ilia Rochlin[4], Ayin Vala[1], Isabella Chu[1], Robert L. Phillips[5], David H. Rehkopf[1,6,7,8,9,10], Neil Kamdar[1,11,12]

1 Center for Population Health Sciences, School of Medicine, Stanford University, Palo Alto, California, United States of America, 2 Division of STD Prevention, Centers for Disease Control and Prevention, Atlanta, Georgia, United States of America, 3 Division of HIV Prevention, Centers for Disease Control and Prevention, Atlanta, Georgia, United States of America, 4 Inform and Disseminate Division, Office of Public Health Data, Surveillance, and Technology, Centers for Disease Control and Prevention, Atlanta, Georgia, United States of America, 5 The Center for Professionalism & Value in Health Care, ABFM Foundation, Seattle, Washington, D.C., United States of America, 6 Department of Epidemiology and Population Health, School of Medicine, Stanford University, Stanford, California, United States of America, 7 Department of Medicine (Division of Primary Care and Population Health), Stanford School of Medicine, Stanford University, Stanford, California, United States of America, 8 Department of Pediatrics, Stanford School of Medicine, Stanford, California, United States of America, 9 Department of Health Policy, Stanford School of Medicine, Stanford University, Stanford, California, United States of America, 10 Department of Sociology, Stanford University, Stanford, California, United States of America, 11 Institute for Healthcare Policy and Innovation, University of Michigan, Ann Arbor, Michigan, United States of America, 12 Cecil G. Sheps Center for Health Services Research, University of North Carolina, Chapel Hill, North Carolina, United States of America

* shiyingh@stanford.edu

## Abstract

### Background

Early identification and treatment of sexually transmitted infections (STIs) is critical to improve patient outcomes. Barriers to healthcare seeking are potentially exacerbated by COVID-19. This study examined trends in STI testing and positivity from 2019 to 2021 in primary care in the United States.

### Methods

This is a retrospective study using the PRIME Registry, a national primary care EHR registry, from January 1, 2019-December 31, 2021. We calculated age-standardized monthly and annual testing rates for chlamydia, gonorrhea, syphilis, and human immunodeficiency virus stratified by gender and race/ethnicity. We also generated quarterly and annual rates for test positivity. Chi-square tests and 95% confidence intervals were used for comparison. 753 practices and 4,410,609 patients were included, with 180,558 having STI tests.

**Data availability statement:** We cannot make the data publicly available due to contractual limitations and the inclusion of sensitive information and PHI. However, individuals wishing to access the data may contact the Stanford Center for Population Health Sciences at phsdatacore@stanford.edu. Upon completion of security, regulatory and contractual requirements, individuals may access the data.

**Funding:** Funder: Centers for Disease Control and Prevention (CDC) Award Number: 75D30122P12974 Grant Recipient: American Board of Family Medicine, Inc. URL: https://www.cdc.gov

**Competing interests:** Neil Kamdar serves as a consultant at the University of North Carolina, Chapel Hill, Sheps Center for Health Policy, and the University of New Mexico, Department of Emergency Medicine. Other authors have no conflicting or competing interests. This does not alter our adherence to PLOS ONE policies on sharing data and materials.

**Abbreviations:** AFC, American Family Cohort; CDC, Centers for Disease Control and Prevention; CI, Confidence Interval; CPT, Current Procedural Terminology; EHR, Electronic Health Records; HCPCS, Healthcare Common Procedure Coding System; HIV, Human Immunodeficiency Virus; ICD, International Classification of Diseases; LOINC, Logical Observation Identifier Names and Codes; RUCA, Rural-urban commuting area; SDI, Social Deprivation Index; STI, Sexually transmitted infection; SVI, Social Vulnerability Index; U.S., United States; USPTF, U.S. Preventative Task Force.

## Results

We observed a substantial decline in testing rates for STIs from March-April 2020 (31% for chlamydia, 30% for gonorrhea, 23% for syphilis, 24% for HIV), followed by a rapid increase in May-June 2020 (64% for chlamydia, 65% for gonorrhea, 32% for syphilis, 48% for HIV). Testing rates per 100,000 decreased from 2019 to 2021 for chlamydia (3,592 vs 2,355 vs 2,181) while increased for gonorrhea in 2020 (2,129 vs 2,207 vs 2,057). STI testing rates from 2019 to 2021 for females and non-Hispanic Black or African American patients were higher than other groups. An increase in test positivity from 2019 to 2021 was observed for gonorrhea (0.4% vs 0.4% vs 0.5%) but no significant change for chlamydia (1.5% vs 1.6% vs 1.5%).

## Conclusion

Testing rates for STIs substantially dropped during stay-at-home orders early in the pandemic and recovered after these were relaxed. Gender and race/ethnicity STI testing differences may reflect primary care's prioritization of higher risk populations. This study emphasizes the role of primary care EHR data in monitoring and an opportunity for closer collaboration with public health agencies.

## Introduction

Cases of sexually transmitted infections (STIs) in the United States (U.S.) rose abruptly over the past decade. From 2014 to 2018, the U.S. experienced a 19% rate increase in chlamydia, a 63% increase in gonorrhea, and a 71% increase in primary and secondary syphilis [1]. Chlamydia, gonorrhea, and syphilis are the most commonly reported STIs in the U.S., and patients with these STIs are often at greater risk of human immunodeficiency virus (HIV) [2,3]. Centers for Disease Control and Prevention (CDC) STI surveillance suggests that STI prevalence had been increasing prior to the COVID-19 pandemic [2]. The growing clinical and social burden of STIs is of public health concern warranting the continued prioritization of routine screening, early detection, and treatment [4].

The global pandemic was associated with disruptions in STI-related care, some of which may have been related to disruptions of usual patterns of primary care [5,6]. Stay-at-home orders were issued in March or April 2020 in the U.S [7]. Societal lockdowns, avoidance of clinical care, and clinical practice closures or other changes in care provision all could have had a detrimental effect on STI screening, diagnosis, and related treatment during the peak of the pandemic [8]. Prior studies supported this risk in the first year of the pandemic [5,8].

Primary care practices are increasing in their role in STI detection and care in recent years and are integral to a patient's utilization of tests and other preventative services [9]. Many healthcare settings are utilized for STI testing in the U.S., including STI clinics, family planning clinics, hospitals, emergency departments, urgent care centers, and physician offices, and differences in how STIs are managed in these

different locations have been noted [10–14]. For the majority of U.S. patients, primary care is the first place to address health needs, including STI-related care [9,15]. Primary care's critical role in screening and diagnosis of STIs in recent years underscores the importance of carefully examining STI tests in this patient care setting [16].

Motivated by declines and barriers to the practical use of primary care with the onset of the COVID-19 pandemic [6] and increasing rates of STIs over the last decade, this study examines STI testing rates and test positivity in the primary care setting. We examined testing rates and test positivity amongst all patients before (2019) and during the COVID-19 pandemic (2020 and 2021). Given existing evidence in variations by gender and race/ethnicity, we examine trends between these groups.

## Materials and methods

### Data source

The study used data from the American Family Cohort (AFC), a research dataset derived from the electronic health records (EHRs) of primary care practices in the PRIME Registry. The PRIME Registry, established in 2016, is a national Centers for Medicare and Medicaid Services Qualified Clinical Data Registry (QCDR) and now includes practices and patients from nearly all states. Primary care practices submit their EHR data to enable production of quality measures and patient care quality dashboards for value-based care reporting requirements and improving care. The registry includes patient demographics, primary care visits, patient problem history, information about procedures and diagnoses, lab tests and associated results, vital signs, medications, clinical notes during the patients' visits, and neighborhood social deprivation indices.

Our study analyzes EHR from January 1, 2019 to December 31, 2021 due to the AFC data availability. Data was accessed for research purposes of this study on October 1, 2022. Identifiable information was included in the original data source, which was then de-identified for subsequent analysis. The study was approved by the Institutional Review Board of Stanford University (No.: 61956).

### Study population–inclusion and exclusion criteria

Practices that had one or more encounters for a test for chlamydia, gonorrhea, syphilis, and HIV during the entire study period were included. A test was identified from an encounter with definitions consistent with the U.S. Preventative Task Force (USPTF) documentation using *International Classification of Diseases (ICD)*, *Current Procedural Terminology* (CPT) [17], *Healthcare Common Procedure Coding System* (HCPCS), *Logical Observation Identifier Names and Codes* (LOINC), or other test codes documented in the registry (S1 Table). Patients that had one or more encounters within any of the included practices were qualified as the study population. Patients with missing information for age were excluded.

### Outcome definition

Our primary outcome was the monthly and annual rates for testing for chlamydia, gonorrhea, syphilis, and HIV during the study period, which were presented as the number of tests per 100,000 patients. The evidence of having a test was shown in S1 Table. Our secondary outcome was the test positivity for chlamydia and gonorrhea, respectively. The evidence of having a positive test was those having results coded as "positive" or "detected" in the EHR. A chlamydia or gonorrhea test could be a test from genital, throat, or rectal areas. A person with positive chlamydia or gonorrhea results of genital, throat, or rectal areas on the same day were counted as one positive chlamydia or gonorrhea test. Test positivity values were presented as the percentage of the tests with positive results documented. The positivity values were calculated quarterly and annually during the study period.

Our primary outcome was calculated as a ratio of the monthly or annual number of the STI tests to the number of "at-risk" patients multiplying 100,000. The numerator of the ratio was the number of the STI tests documented in the AFC

EHR within that month or year. The calculation of the denominator of the ratio was broken down into two steps. In Step 1, we identified the active primary care practices at the beginning of the month or year, which were practices having (1) one or more encounters within that month or year, and (2) one or more encounters for STI tests during the entire study period. We required this definition to identify the primary care practices addressing patient STI concerns. In Step 2, we defined the at-risk patient population based on a patient that had one or more encounters within any of the active primary care practices identified in Step 1 within that month or year or within the prior 12-month period. We excluded subsequent patient-months after the month of death for the entire study period. We required this definition to ensure that we have capture of the at-risk patient population that is eligible for testing in the patient-month and/or patient-year.

Our secondary outcome was calculated as a ratio of those having positive results to tests having results captured by the primary care practices within that quarter or year. The numerator of the ratio was the number of tests with positive test results documented in the AFC EHR. The denominator of the ratio was the number of tests having evidence of documented results in the AFC EHR. The tests performed at primary care practices that do not have documented results were excluded from the denominator since practices in the PRIME Registry may have evidence of tests performed but not necessarily the test results. We used these conditions to ensure proper documentation of test results for any given patient calendar quarter and year.

## Patient characteristics

Patient characteristics studied included gender, age group at the beginning of 2019, race/ethnicity, U.S. Census Region, Social Deprivation Index (SDI), [18] Social Vulnerability Index (SVI), [19] and Rural-urban commuting area (RUCA) codes [20]. Age groups were categorized into increments of 10-years for descriptive analysis (**Table 1**). Race/ethnicity were categorized into American Indian or Alaska Native, Asian, non-Hispanic Black or African American, Hispanic or Latino, Native Hawaiian or Other Pacific Islander, non-Hispanic White, Multiple Races, or Other/Unknown race. SDI, SVI, and RUCA were associated with patients based on the zip codes of patient addresses captured in the EHR [18–20].

## Statistical analysis

We summarized baseline demographic characteristics for all patients and further stratified the demographic characteristics by specific STI testing during the study period. We performed Chi-Square tests to determine whether differences observed were likely due to chance.

We applied the standardization to the monthly and annual rates of STI tests. Direct age adjustment was based on using the reference U.S. age distribution per the U.S. Census 2020 data for the study period. This is based on Annual Estimates of the Resident Population by Sex, Age, Race, and Hispanic Origin for the United States: April 1, 2020 to July 1, 2021 [21].

Monthly and annual rates of STI tests were calculated and presented for all patients and by category (gender: female and male; race/ethnicity: non-Hispanic Whites, non-Hispanic Blacks or African Americans, Hispanic/Latino). Gender-specific and race/ethnicity-specific testing rates for STIs were age and gender standardized and age and race/ethnicity standardized, respectively, based on direct age adjustment using the age and gender distribution and age and race/ethnicity distribution of the U.S. Census 2020 data [21]. The 95% confidence interval (CI) was calculated based on the standard errors generated using the Keyfitz formula:[22]

$$95\% \ CI = \ \pm 1.96 \times \frac{R}{\sqrt{N}}$$

where R is adjusted rates for STI testing, N is number of tests.

**Table 1. Characteristics of patients who had STI tests, American Family Cohort, 2019-2021.**

| Characteristics | Chlamydia tests[a], N=132,076 | Gonorrhea tests, N=86,672 | Syphilis tests, N=69,336 | HIV tests, N=90,083 | No record of STI testing, N=4,230,051 | Overall, N=4,410,609 |
|---|---|---|---|---|---|---|
| Gender, N (%) | | | | | | |
| Female | 98,277 (74.4) | 61,939 (71.5) | 45,319 (65.4) | 57,049 (63.3) | 2,319,569 (54.8) | 2,445,682 (55.4) |
| Male | 33,786 (25.6) | 24,705 (28.5) | 23,985 (34.6) | 33,023 (36.7) | 1,907,827 (45.1) | 1,962,230 (44.5) |
| Other/Unknown | 13 (0.01) | 28 (0.03) | 32 (0.05) | 11 (0.01) | 2,655 (0.1) | 2,697 (0.1) |
| Age in years on 01/01/2019, N (%) | | | | | | |
| <10 | 276 (0.2) | 213 (0.2) | 112 (0.2) | 104 (0.1) | 334,975 (7.9) | 335,447 (7.6) |
| 10-19 | 23,322 (17.7) | 15,113 (17.4) | 8,247 (11.9) | 10,896 (12.1) | 381,167 (9) | 408,537 (9.3) |
| 20-29 | 48,670 (36.8) | 32,626 (37.6) | 26,228 (37.8) | 31,393 (34.8) | 454,059 (10.7) | 513,539 (11.6) |
| 30-39 | 28,560 (21.6) | 19,318 (22.3) | 16,234 (23.4) | 20,731 (23) | 512,274 (12.1) | 551,215 (12.5) |
| 40-49 | 15,966 (12.1) | 10,351 (11.9) | 7,971 (11.5) | 11,743 (13) | 558,072 (13.2) | 581,803 (13.2) |
| 50-59 | 9,433 (7.1) | 5,735 (6.6) | 4,952 (7.1) | 8,448 (9.4) | 669,582 (15.8) | 686,030 (15.6) |
| 60-69 | 4,226 (3.2) | 2,491 (2.9) | 3,009 (4.3) | 4,807 (5.3) | 657,974 (15.6) | 667,073 (15.1) |
| 70+ | 1,623 (1.2) | 825 (1) | 2,583 (3.7) | 1,961 (2.2) | 661,948 (15.6) | 666,965 (15.1) |
| Race/ethnicity, N (%) | | | | | | |
| American Indian or Alaska Native | 336 (0.3) | 253 (0.3) | 129 (0.2) | 2,773 (3.1) | 30,954 (0.7) | 33,951 (0.8) |
| Asian | 2,491 (1.9) | 1,831 (2.1) | 1,362 (2) | 1,775 (2) | 84,487 (2) | 88,264 (2) |
| Non-Hispanic Black or African American | 22,413 (17) | 13,672 (15.8) | 11,607 (16.7) | 14,135 (15.7) | 276,151 (6.5) | 303,282 (6.9) |
| Hispanic or Latino | 19,442 (14.7) | 15,118 (17.4) | 11,830 (17.1) | 15,935 (17.7) | 402,061 (9.5) | 430,990 (9.8) |
| Multiple Races | 53 (0.04) | 13 (0.01) | 11 (0.02) | 4 (0) | 1,530 (0.04) | 1,591 (0.04) |
| Native Hawaiian or Other Pacific Islander | 200 (0.2) | 130 (0.1) | 79 (0.1) | 100 (0.1) | 7,822 (0.2) | 8,072 (0.2) |
| Non-Hispanic White | 62,791 (47.5) | 38,478 (44.4) | 29,334 (42.3) | 37,484 (41.6) | 2,577,282 (60.9) | 2,659,469 (60.3) |
| Other/Unknown | 24,350 (18.4) | 17,177 (19.8) | 14,984 (21.6) | 17,877 (19.8) | 849,764 (20.1) | 884,990 (20.1) |
| US Census Region, N (%) | | | | | | |
| Midwest | 38,364 (29) | 26,816 (30.9) | 21,108 (30.4) | 23,805 (26.4) | 765,080 (18.1) | 809,965 (18.4) |
| Northeast | 19,549 (14.8) | 5,705 (6.6) | 5,401 (7.8) | 7,955 (8.8) | 405,233 (9.6) | 429,710 (9.7) |
| South | 50,105 (37.9) | 34,507 (39.8) | 28,440 (41) | 36,996 (41.1) | 2,228,902 (52.7) | 2,302,141 (52.2) |
| West | 23,905 (18.1) | 19,592 (22.6) | 14,301 (20.6) | 18,618 (20.7) | 780,398 (18.4) | 815,512 (18.5) |
| Unknown | 153 (0.1) | 52 (0.1) | 86 (0.1) | 2,709 (3) | 50,438 (1.2) | 53,281 (1.2) |
| Social Deprivation, median (IQR) | | | | | | |
| Social Deprivation Index (SDI) | 52 (31, 73) | 57 (30, 79) | 57 (30, 77) | 57 (29, 82) | 47 (22, 70) | 48 (22, 70) |
| Social Vulnerability Index (SVI) | 0.5 (0.4, 0.7) | 0.6 (0.4, 0.8) | 0.6 (0.4, 0.8) | 0.6 (0.4, 0.8) | 0.5 (0.3, 0.8) | 0.5 (0.3, 0.8) |
| Rural-urban commuting area (RUCA), N (%) | | | | | | |
| Metropolitan | 99,186 (75.1) | 62,910 (72.6) | 53,220 (76.8) | 66,211 (73.5) | 2,885,894 (68.2) | 3,020,592 (68.5) |
| Micropolitan | 16,311 (12.3) | 12,296 (14.2) | 8,214 (11.8) | 11,466 (12.7) | 692,204 (16.4) | 714,997 (16.2) |
| Small town | 12,235 (9.3) | 9,092 (10.5) | 5,953 (8.6) | 7,327 (8.1) | 391,344 (9.3) | 406,030 (9.2) |
| Rural | 4,191 (3.2) | 2,323 (2.7) | 1,864 (2.7) | 2,371 (2.6) | 210,025 (5) | 215,563 (4.9) |
| Unknown | 153 (0.1) | 51 (0.1) | 85 (0.1) | 2,708 (3) | 50,584 (1.2) | 53,427 (1.2) |

[a] Chlamydia and gonorrhea tests are often ordered together in primary care. However, in our data, a few practices documented chlamydia tests in the EHR but didn't document any gonorrhea tests throughout the entire study period, resulting in a difference in total counts between the two tests. The absence of gonorrhea test documents from these practices could be due to variations in EHR data quality and integrity, which limits our access to them.

Changes in STI testing rates from pre- (2019) to during-pandemic (2020, 2021) periods were assessed by both the changes in monthly rates and annual rates. Furthermore, monthly rates during 2020 and 2021 were compared to the monthly rates during 2019 in the same months to account for seasonal differences.

We compared testing rates between months and years and between race/ethnicity and gender groups using the 95% CIs. We compared test positivity between years and quarters using Chi-Square tests.

All the analyses were performed in Python (Version 3.7.10). All significance testing was at the 0.05 significance level or 95% confidence level.

## Results

### Baseline characteristics

A total of 753 primary care practices (S1 Fig) and 4,410,609 patients were included in the study. Patients had a total of 233,510 tests for chlamydia, 135,559 tests for gonorrhea, 109,909 tests for syphilis, and 135,953 tests for HIV in 2019–2021. Among these patients, 180,558 had one or more STI tests (chlamydia: 132,076 patients; 702 practices; gonorrhea: 86,672 patients; 438 practices; syphilis: 69,336 patients; 475 practices; HIV: 90,083 patients; 552 practices) in 2019–2021. A significantly higher proportion of females, patients aged 10–39 years old, patients who identified as non-Hispanic Black or African American, or Hispanic/Latino were tested for STIs compared to their counterparts (P < 0.001, Chi-squared test; **Table 1**). Patients aged 10–24 consistently had higher testing rates (P < 0.001, Chi-squared test) than others in 2020 and 2021 (chlamydia: 4% vs 1% in 2020 and 4% vs 0.9% in 2021; gonorrhea: 4% vs 1% in 2020 and 4% vs 0.9% in 2021; syphilis: 2% vs 0.8% in 2020 and 2% vs 0.7% in 2021; HIV: 3% vs 1% in both 2020 and 2021). Gender and race/ethnicity information was missing for round 0.1% and 20% of the population, respectively.

### Trends of STI tests

A substantial drop in STI testing rates was observed for all four STIs from January 2020 to April 2020 (chlamydia: 53% decline; from 346 [95% CI: 338–353] to 163 [95% CI: 158–168] tests per 100,000 patients; gonorrhea: 46%; from 285 [95% CI: 278–293] to 153 [95% CI: 147–159] tests per 100,000 patients; syphilis: 36%; from 209 [95% CI: 202–215] to 133 [95% CI: 128–138] tests per 100,000 patients; HIV: 42%; from 235 [95% CI: 229–242] to 137 [95% CI: 132–143] tests per 100,000 patients) (**Fig 1**, Table 2). Specifically, a drop from March 2020 to April 2020 (31% for chlamydia, 30% for gonorrhea, 23% for syphilis, 24% for HIV) were observed. This early-pandemic drop rebounded from April 2020 to June 2020 but not to pre-pandemic volume (chlamydia: 64% increase; 163 [95% CI: 158–168] to 267 [95% CI: 260–274] tests per 100,000 patients; gonorrhea: 65%; from 153 [95% CI: 147–159] to 253 [95% CI: 246–261] tests per 100,000 patients; syphilis: 42%; from 133 [95% CI: 128–138] to 189 [95% CI: 181–193] tests per 100,000 patients; HIV: 48%; from 137 [95% CI: 132–143] to 203 [95% CI: 196–209] tests per 100,000 patients).

Rates of all four STI tests further decreased since August 2021 (**Fig 1**). From August to September 2021, testing rates dropped by 30% (from 267 [95% CI: 259–275] to 187 [95% CI: 181–194] per 100,000 patients) for chlamydia, by 27% (from 231 [95% CI: 224–240] to 169 [95% CI: 162–176] per 100,000 patients) for gonorrhea, by 29% (from 168 [95% CI: 162–176] to 119 [95% CI: 114–125] per 100,000 patients) for syphilis, and by 20% (from 208 [95% CI: 201–215] to 166 [95% CI: 159–172] per 100,000 patients) for HIV.

The annual rates of chlamydia tests in 2020 (2,355 [95% CI: 2,337−2,373] per 100,000 patients) and 2021 (2,181 [95% CI: 2,162−2,200] per 100,000 patients) were lower than in 2019 (3,592 [95% CI: 3,572−3,613] per 100,000 patients), while the annual rate of gonorrhea tests in 2020 (2,207 [95% CI: 2,187−2,226] per 100,000 patients) was higher than in 2019 (2,129 [95% CI: 2,110−2,147] per 100,000 patients) (S2 Table). Monthly patterns (**Fig 2**) illustrate that chlamydia testing rates are consistently lower in 2020 and 2021 across all calendar months compared to 2019. However, gonorrhea testing rates remained higher from June 2020 – July 2021 than in the same period of 2019 and then decreased thereafter.

### Test positivity

Annual test positivity for chlamydia in 2020 (1.6%; 1,001 of 63,569) and 2021 (1.5%; 699 of 45,686) were like 2019 (1.5%; 1,715 of 110,294). For gonorrhea, there is a higher test positivity (P = 0.03) for 2021 (0.5%; 147 of 31,737) compared to

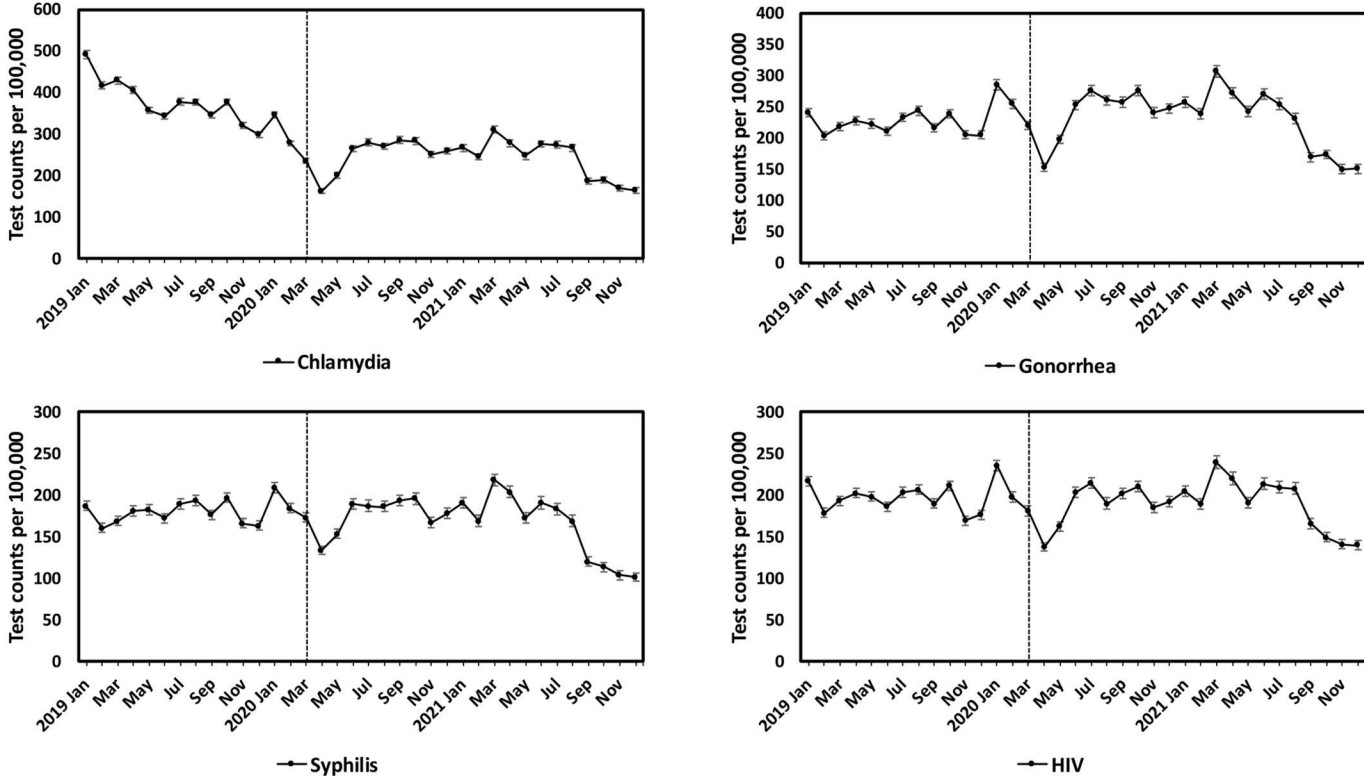

**Fig 1. Monthly number of tests per 100,000 patients for chlamydia, gonorrhea, syphilis, and HIV in the American Family Cohort from 2019 to 2021, age standardized to the U.S. standard population.** Error bars show the 95% confidence intervals. The vertical dashed line represents the beginning of the stay-at-home order in March 2020.

2019 (0.4%; 163 of 45,519), while there was no significant change (P = 0.06) from 2019 to 2020 (0.4%; 192 of 43,643). Quarterly patterns illustrate that chlamydia had a higher test positivity (P < 0.05) for Q2-Q4 for 2020 (1.6%; 717 of 43,964) and 2021 (1.6%; 487 of 31,415) compared to 2019 (1.4%; 1,050 of 77,224); however, Q1 2019 (2.0%; 665 of 33,070) had a substantially higher test positivity (P < 0.001) compared to Q1 2020 (1.4%; 284 of 19,605) and 2021 (1.5%; 212 of 14,271), respectively (**Fig 3**). Quarterly patterns also illustrate that gonorrhea had a higher test positivity (P < 0.05) for Q2-Q4 2020 (0.5%; 152 of 30,962) and 2021 (0.5%; 102 of 21,350) compared to 2019 (0.3%; 119 of 34,687), respectively. The test positivity for gonorrhea was trending higher for Q1 2021 compared to 2019 but not statistically significant.

## Gender and race/ethnicity variations in STI tests

STI testing rates were higher in females than males in 2019–2021 (**Fig 4**). By excluding primary care visits from seven women-specific practices, males had higher HIV and syphilis testing rates than females (**S2 Fig**). While the testing rates for chlamydia in 2020 were lower than in 2019 for both males and females, the annual testing rate for gonorrhea in 2020 increased by 10% among females (2020 vs. 2019: 2,736 [95% CI: 2,707−2,756] vs. 2,488 [95% CI: 2,462−2,513] per 100,000 patients) but decreased by 12% among males (2020 vs. 2019: 1,406 [95% CI: 1,382−1,431] vs. 1,592 [95% CI: 1,567−1,616] per 100,000 patients).

Non-Hispanic Black or African American patients had higher testing rates for all four STIs than Hispanics and non-Hispanic whites in 2019–2021 (**Fig 5**, **S2 Table**). For gonorrhea, there were obvious differences between the three racial/

**Table 2. Monthly number of tests per 100,000 patients and the 95% confidence intervals (CIs) for the chlamydia, gonorrhea, syphilis, and HIV in the American Family Cohort from 2019 to 2021.**

| Test counts per 100,000 (95% CI) | 2019 | 2020 | 2021 |
|---|---|---|---|
| Chlamydia | | | |
| January | 492 (483-500) | 346 (338-353) | 267 (259-274) |
| February | 417 (410-425) | 278 (271-285) | 245 (238-252) |
| March | 430 (422-437) | 235 (229-242) | 311 (303-319) |
| April | 406 (398-414) | 163 (158-168) | 278 (270-286) |
| May | 358 (351-366) | 201 (195-207) | 247 (240-255) |
| June | 343 (336-350) | 267 (260-274) | 276 (268-284) |
| July | 378 (371-386) | 280 (273-288) | 275 (267-283) |
| August | 377 (370-385) | 272 (265-279) | 267 (259-275) |
| September | 346 (339-353) | 286 (279-293) | 187 (181-194) |
| October | 377 (369-384) | 284 (276-291) | 190 (184-197) |
| November | 320 (313-327) | 250 (243-257) | 170 (163-176) |
| December | 299 (292-306) | 260 (253-267) | 165 (158-171) |
| Gonorrhea | | | |
| January | 241 (234-248) | 285 (278-293) | 258 (250-266) |
| February | 204 (198-210) | 255 (248-262) | 239 (231-248) |
| March | 219 (213-226) | 220 (213-227) | 308 (299-317) |
| April | 228 (222-235) | 153 (147-159) | 273 (264-282) |
| May | 223 (216-230) | 198 (191-205) | 242 (234-250) |
| June | 211 (205-218) | 253 (246-261) | 271 (262-280) |
| July | 234 (227-240) | 276 (268-284) | 255 (246-263) |
| August | 244 (237-251) | 260 (253-268) | 232 (224-240) |
| September | 217 (210-224) | 257 (249-265) | 169 (162-176) |
| October | 239 (232-246) | 276 (268-284) | 174 (166-181) |
| November | 205 (199-212) | 241 (233-249) | 150 (144-157) |
| December | 205 (198-211) | 248 (240-256) | 151 (144-158) |
| Syphilis | | | |
| January | 187 (181-193) | 209 (202-215) | 191 (184-198) |
| February | 161 (155-166) | 184 (178-190) | 169 (162-176) |
| March | 169 (163-175) | 173 (167-179) | 218 (211-226) |
| April | 181 (175-187) | 133 (128-138) | 204 (196-211) |
| May | 183 (177-188) | 153 (148-159) | 173 (166-179) |
| June | 172 (166-178) | 189 (182-195) | 191 (184-198) |
| July | 189 (183-195) | 187 (181-193) | 183 (176-190) |
| August | 193 (187-199) | 187 (180-193) | 169 (162-176) |
| September | 176 (170-182) | 193 (187-200) | 120 (114-125) |
| October | 197 (191-203) | 196 (189-202) | 114 (108-119) |
| November | 166 (160-171) | 167 (160-173) | 104 (98-109) |
| December | 163 (158-169) | 178 (172-185) | 101 (96-107) |
| HIV | | | |
| January | 217 (211-223) | 235 (229-242) | 205 (198-211) |
| February | 179 (173-184) | 198 (192-204) | 189 (182-196) |
| March | 193 (188-199) | 181 (175-187) | 239 (232-247) |
| April | 202 (196-208) | 137 (132-143) | 220 (213-227) |
| May | 198 (192-204) | 162 (157-168) | 191 (184-197) |

*(Continued)*

**Table 2.** (Continued)

| Test counts per 100,000 (95% CI) | 2019 | 2020 | 2021 |
|---|---|---|---|
| June | 186 (181-192) | 203 (196-209) | 214 (207-221) |
| July | 203 (198-209) | 215 (208-221) | 209 (202-216) |
| August | 207 (201-212) | 190 (184-196) | 208 (201-215) |
| September | 190 (184-195) | 202 (196-209) | 166 (159-172) |
| October | 211 (205-217) | 210 (204-217) | 149 (143-155) |
| November | 169 (164-175) | 185 (179-191) | 141 (135-147) |
| December | 176 (171-182) | 192 (186-199) | 140 (134-146) |

ethnic groups' testing rates in 2020 and 2021 compared to 2019. Non-Hispanic Black or African American patients had higher annual gonorrhea testing rates in 2020 (6,802 [95% CI: 6,669−6,936] per 100,000 patients) and 2021 (6,820 [95% CI: 6,670−6,970] per 100,000 patients) than in 2019 (5,300 [95% CI: 5,189−5,411] per 100,000 patients). Conversely, Hispanic patients had lower testing rates in 2020 and 2021 than in 2019, and non-Hispanic white patients only had lower testing rates in 2021 than in 2019. (S2 Table).

## Discussion

This study investigated the testing rates for chlamydia, gonorrhea, syphilis, and HIV from 2019 to 2021 from 753 primary care practices, including 4,410,609 patients across the U.S. In 2020, during the initial phase of the pandemic, all the STI testing rates experienced drastic declines from January to April before a substantial rebound in May and June. Unlike chlamydia, gonorrhea testing rates increased in 2020 compared to 2019. Females had higher annual STI testing rates than males, and non-Hispanic Black or African American patients had higher annual testing rates than Hispanic patients and non-Hispanic white patients from 2019 to 2021.

These findings demonstrate there was a substantial disruption in STI testing during the COVID-19 pandemic year 2020 with lingering effects in some cases in 2021. Some of these disruptions may be attributed to the stay-at-home orders issued in March or April 2020 [23]. There was also a decline prior to the stay-at-home order which could be related to a general public concern about news of a potential pandemic beginning in January 2020 [24]. This could have resulted in fewer patients seeking primary care for STI screening in the months immediately antecedent to the pandemic period [25]. Furthermore, COVID-19 case incidence increased in late 2021 (August-December),[26] and several states announced or extended state of emergency in response to a rising number of COVID-19 cases and hospitalizations,[27–31] which aligns with a reduction in the STI testing rates during the same time period. Relatedly, the increase in the case incidence as well as the various lockdowns during the pandemic period had led to a decrease in primary care practices remaining open or their overall capacity to manage the same patient volumes as they did during the pre-pandemic period [23]. Therefore, patients' access to primary care during the pandemic period had been negatively affected leading to increased barriers in receiving STI testing.

Gonorrhea and chlamydia testing trends in these primary care practices were consistent with general trends in CDC reports on the case incidence [8]. Such trends may be explained by the updated public health activity guidance in response of the COVID-19 public emergency [32]. The fact that the stay-at home order ended in many states since May 2020 may also have contributed to the testing rate increase later in the year.[7] This enabled primary care practices to resume services and perhaps increase STI testing due to increased demand post initial COVID-19 phase [33]. Following the CDC's guidance, syphilis and gonorrhea were made a top priority by many health departments. This might explain the significant decline in chlamydia tests, which was potentially due to the limited resources, and the increase in gonorrhea tests for all patients and the increase in syphilis tests for certain populations in late 2020, compared to their testing rates in the same period in 2019 [8]. Notably, some US-based studies reported a decrease in gonorrhea testing volumes in 2020

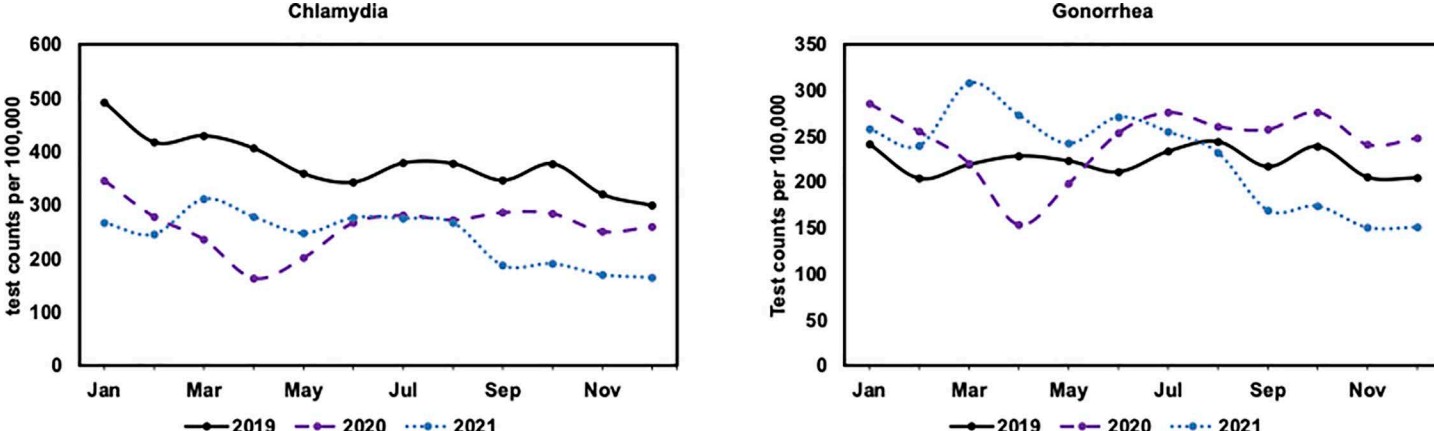

**Fig 2. Monthly number of tests per 100,000 patients for chlamydia (left) and gonorrhea (right) tests in the American Family Cohort in 2019, 2020, and 2021, age standardized to the U.S. standard population.**

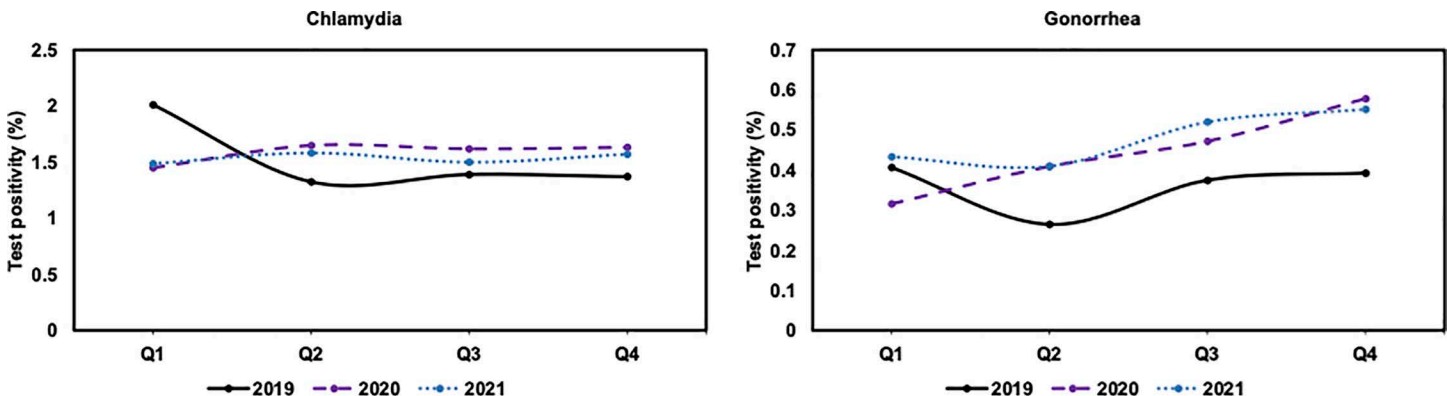

**Fig 3. Quarterly test positivity for chlamydia (left) and gonorrhea (right) in the American Family Cohort from 2019 to 2021.**

compared to 2019 [33,34]. This might reflect variations in resource allocation patterns across different clinical settings during the pandemic. This might also indicate an improvement in gonorrhea test documentation within structured EHR systems in PRIME practices, which could result in higher observed gonorrhea testing rates.

Our findings that females and non-Hispanic Black or African American patients having higher annual STI testing rates during our study period are consistent with the results of a few studies [33–37]. This may be explained by several phenomena. The high rate amongst females could be due to the U.S. Preventive Services Task Force (USPSTF) recommendations for STI screening for sexually active young women or due to pregnancy screening [38]. However, there is insufficient evidence in our study to suggest if STI testing is screening or diagnosis. Studies based on EHR, surveys, and claims data also found that Black or African American patients were more likely to receive STI tests compared to Whites [35–37,39]. This is potentially related to the finding that this population were seen at a higher risk of having STIs and related conditions than White patients, [2,40] which might have led to a risk-based prioritization in providing STI testing services [39]. Further, research within a U.S. hospital system found that despite a reduction in overall STI testing counts during the early pandemic, females and non-Hispanic Black or African American

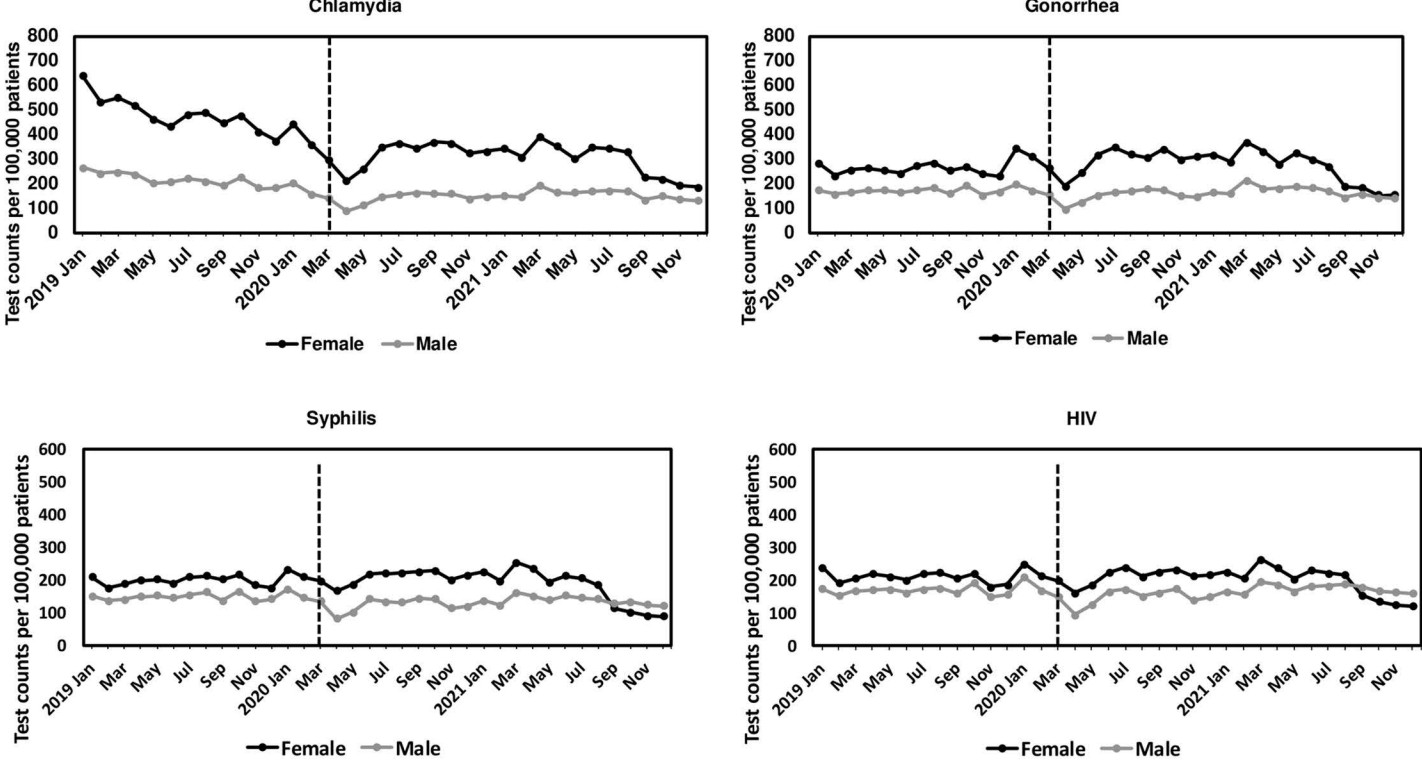

**Fig 4. Monthly number of tests per 100,000 patients for chlamydia, gonorrhea, syphilis, and HIV in the American Family Cohort from 2019 to 2021 by gender, age standardized to the U.S. standard population.** The vertical dashed line represents the beginning of the stay-at-home order in March 2020.

patients had increased testing proportions compared to males and Whites [33]. These findings, together with ours, may reflect that the clinical settings prioritized certain populations for STI testing in response to the public health emergency [5].

This study has limitations that should be considered. STI testing trends in 2022 were not investigated. More recent data would enable understanding testing in the latter part of the pandemic. Patients receive STI testing outside primary care practices, therefore, AFC may not capture testing or result data for some patients. Primary care providers' data documented in AFC does not provide information regarding the process regarding how tests are ordered. Another potential limitation is that tests were identified using procedure codes; therefore, tests that were documented in free-text fields were not captured in this study. Furthermore, we do not know symptom presentation at the time of testing from the procedure codes; therefore, we are unable to discern tests for screening versus diagnosis or investigate how symptom presentation influenced the relationship between patient-level characteristics and testing rates. Additionally, given that a large proportion of chlamydia and gonorrhea cases are asymptomatic, [41–43] routine screening for STIs is essential for early identification and treatment [41,43]. We are unable to examine the proportion of routine screening, though. We are also unable to access whether clinics prioritized tests to symptomatic patients due to limited resources during the COVID [33,34]. Incomplete data was also a limitation. Sexual orientation and gender diversity information was not available in our data, and race/ ethnicity information was missing in approximately 20% of patients in the study. More accurate race/ethnicity groups may be needed in future work via race imputation techniques. Pregnancy-related data were not collected in this

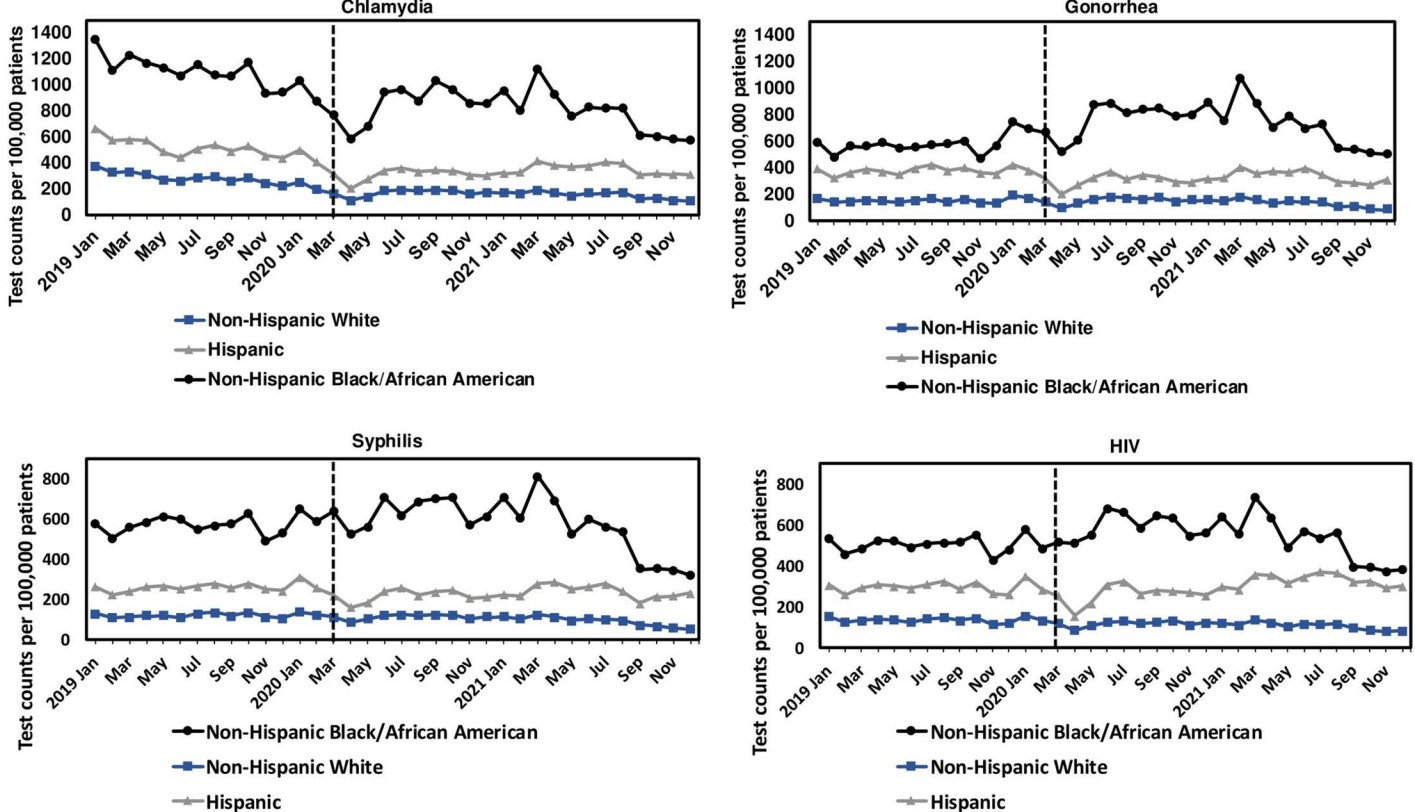

**Fig 5. Monthly number of tests per 100,000 patients for chlamydia, gonorrhea, syphilis, and HIV in the American Family Cohort from 2019 to 2021 by race/ethnicity, age standardized to the U.S. standard population.** The vertical dashed line represents the beginning of the stay-at-home order in March 2020.

study. The role of pregnancy-related screening in testing rates and the adherence to USPSTF and CDC's recommendation [38,44,45], therefore, is unknown.

## Conclusion

Pandemics or other disruptions of normal public functions can interrupt monitoring and treating STIs. This study investigated the monthly and annual testing rates for chlamydia, gonorrhea, syphilis, and HIV in the U.S. primary care practices. Testing rates substantially dropped during stay-at-home orders early in the pandemic and recovered after these were relaxed. Gender and race/ethnicity STI testing differences may reflect primary care's prioritization of higher risk populations and pregnancy-related testing guidance. This study emphasizes the role of primary care EHR data in monitoring and an opportunity for closer collaboration with public health agencies.

## Supporting information

**S1 Table. Procedure codes used to identify the tests for chlamydia, gonorrhea, syphilis, and HIV.** ICD, International Classification of Diseases; CPT, Current Procedural Terminology; HCPCS, Healthcare Common Procedure Coding System. LOINC: Logical Observation Identifiers, Names and Codes.
(PDF)

**S2 Table. Annual number of tests per 100,000 patients and the 95% confidence intervals (CIs) for the chlamydia, gonorrhea, syphilis, and HIV in the American Family Cohort from 2019 to 2021.**
(PDF)

**S1 Fig. Distribution of the 753 primary care practices by U.S. states.**
(PDF)

**S2 Fig. Monthly number of tests per 100,000 patients for chlamydia, gonorrhea, syphilis, and HIV in the American Family Cohort from 2019 to 2021 by gender, age standardized to the U.S. standard population.** Primary care visits from seven women-specific practices excluded. The vertical dashed line represents the beginning of the stay-at-home order in March 2020.
(PDF)

## Acknowledgments

The authors thank Lusilda Agolli, MHA, for her support in formatting the paper.

## Author contributions

**Conceptualization:** William S. Pearson, Ilia Rochlin.

**Data curation:** Ayin Vala.

**Formal analysis:** Shiying Hao.

**Methodology:** Shiying Hao, Ilia Rochlin, David H. Rehkopf.

**Project administration:** Isabella Chu.

**Resources:** Esther E. Velásquez, William S. Pearson, Karen W. Hoover, Weiming Zhu, Isabella Chu, Robert L. Phillips.

**Software:** Esther E. Velásquez, Ayin Vala.

**Supervision:** David H. Rehkopf, Neil Kamdar.

**Validation:** William S. Pearson, Weiming Zhu.

**Visualization:** Shiying Hao.

**Writing – original draft:** Shiying Hao, Neil Kamdar.

**Writing – review & editing:** Esther E. Velásquez, William S. Pearson, Karen W. Hoover, Weiming Zhu, Ilia Rochlin, Robert L. Phillips, David H. Rehkopf.

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
