## [Decision Letter · Decision Letter 0]

9 May 2024

PONE-D-24-07258Primary care usage patterns associated with sexually transmitted infections during COVID-19PLOS ONE

Dear Dr. Hao,

Thank you for submitting your manuscript to PLOS ONE. After careful consideration, we feel that it has merit but does not fully meet PLOS ONE’s publication criteria as it currently stands. Therefore, we invite you to submit a revised version of the manuscript that addresses the points raised during the review process.

Please fully address all reviewer comments.

We look forward to receiving your revised manuscript.

Kind regards,

Janet E Rosenbaum, Ph.D.

Academic Editor

PLOS ONE

“Funder: Centers for Disease Control and Prevention (CDC)

Award Number: 75D30122P12974

Grant Recipient: American Board of Family Medicine, Inc.

URL: https://www.cdc.gov”

“Neil Kamdar serves as a consultant at the University of North Carolina, Chapel Hill, Sheps Center for Health Policy, and the University of New Mexico, Department of Emergency Medicine. Other authors have no conflicting or competing interests.”

5. We notice that your supplementary figures and tables are included in the manuscript file. Please remove them and upload them with the file type 'Supporting Information'. Please ensure that each Supporting Information file has a legend listed in the manuscript after the references list.

Reviewers' comments:

Reviewer's Responses to Questions

**Comments to the Author**

1. Is the manuscript technically sound, and do the data support the conclusions?

Reviewer #1: Partly

Reviewer #2: Partly

2. Has the statistical analysis been performed appropriately and rigorously? 

Reviewer #1: No

Reviewer #2: Yes

3. Have the authors made all data underlying the findings in their manuscript fully available?

Reviewer #1: No

Reviewer #2: Yes

4. Is the manuscript presented in an intelligible fashion and written in standard English?

Reviewer #1: Yes

Reviewer #2: Yes

5. Review Comments to the Author

Reviewer #1: My concerns include the following:

1. Chlamydia (CT) and gonorrhea (GC) can be (and should be) done with swab specimen or urine, so it's not clear why the GC tests were significantly greater than the CT tests. A provider may only order one of the tests but to have such a big difference makes me question the ordering process.

2. CT and GC can be in the vagina/cervix, penis, throat, and/or rectum. It's possible that one patient had a syphilis test, an HIV test, a throat CT/GC test, a rectal CT/GC test, and a urine CT/GC test at the same visit. It's not clear in the manuscript if this was accounted for (extragenital site vs genital site CT/GC tests). And for the positive rates, would a person with a positive GC result of the throat and the rectum on the same day we counted as 1 positive GC or 2 positive GC tests?

3. Why were syphilis and HIV positivity rates not done, and not mentioned in the abstract results? Syphilis is complicated I understand since a positive/reactive test result doesn't necessarily mean an untreated or "new" infection.

4. What were the outcomes of the age groups? It would be informative to know the p-value of age group and testing outcomes. Adolescents and young adults have the highest rates of bacterial STIs but were they more or less likely to be tested during COVID?

5. Department of Health STI clinics are a common place for people to go for STI testing since it is a public clinic (and testing and treatment is free if not insured) along with family planning clinics - neither of these types of clinics were mentioned on page 5-6 when listing U.S. STI testing site types.

6. Several typos: Human Immunodeficiency Virus should be human immunodeficiency virus on page 5; on abbreviations, you have STI and STIs - I would just use STIs or remove the "s" on infections in the STI write out; Reference #2 has a broken link; page 8, "sexr" should be "sex".

7. Gay, bisexual, and other MSM may be the reason more "men" were tested for HIV and syphilis compared to women. However, during this time period, and worse now, congenital syphilis rates were increasing, so women should be being tested along with heterosexual men, but none of this is mentioned. I'm assuming there is no sexual orientation identity demographic data in the data set used, but this should be listed as a limitation.

Reviewer #2: Reviewer comments

Thank you for considering me as a reviewer for the study entitled: “Primary care usage patterns associated with sexually transmitted infections during COVID-19” which I read with great interest.

The authors examined trends in STI testing and positivity before and during the COVID-19

pandemic in the primary care setting using retrospective data from the PRIME Registry collected from January 1, 2019-December 31, 2021.

Several questions remain that need further clarification please see below.

Abstract

Methods

“We calculated age-standardized monthly and annual STI testing rates and stratified by gender and race and ethnicity.”

Reviewer: I think it would help to clarify which STI you examined.

“We calculated age-standardized monthly and annual STI testing rates and stratified by sex and race and ethnicity. We also generated quarterly and annual rates for test positivity.”

Reviewer: What was the rational to calculate monthly and annual STI testing rates but for test positivity quarterly and annual rates?

Introduction

Reviewer: I think it would be good for the understanding of the results to include information on times of lockdowns in the US.

Methods

Reviewer: Based on what did you identify positive test results, how are they coded in the AFC EHR? How likely is miscoding? In S2 Appendix you provided the codes to identify tests, I think it would be good to provide the codes used to identify positive tests. Using for example ICD-10 codes for HIV can be tricky due to miscoding.

Reviewer: Did you distinguish between different stages of Syphilis and latent and acute Syphilis?

“Patients with missing information for age were excluded.”

Reviewer: I assume this was because you used age for the standardization?

“Our primary outcome was the monthly and annual rates for testing for chlamydia, gonorrhea, syphilis, and HIV during the study period, …”

“Our secondary outcome was the test positivity for chlamydia and gonorrhea, respectively.”

Reviewer: Why did you only look at chlamydia and gonorrhea for the secondary outcome?

“Monthly and annual rates of STI tests were calculated and presented for all patients and by category (sex: female and male; race and ethnicity: non-Hispanic Whites, non-Hispanic Blacks or African Americans, Hispanic/Latino).”

Reviewer: Does the data only provide female or male sex, what about sex divers?

Results

Reviewer: To avoid confusion I would reorder the sentences and already include the total number of tests in the first paragraph as follows:

A total of 753 primary care practices (S4 Appendix) and 4,410,609 patients were included in the study. Patients had a total of 233,510 tests for chlamydia, 135,559 tests for gonorrhea, 109,909 tests for syphilis, and 135,953 tests for HIV in 2019 – 2021. Among these patients, 180,558 had one or more STI tests (chlamydia: 132,076 patients; 702 practices; gonorrhea: 86,672 patients; 438 practices; syphilis: 69,336 patients; 475 practices; HIV: 90,083 patients; 552 practices) in 2019 – 2021.

Trends of STI tests

Reviewer: I would think about including the information into a table as there are a lot of numbers and CI presented and a table would help to compare the different rates over time.

“Rates of all four STI tests further decreased since August 2021 (Fig1).”

Reviewer: See comment on lockdown times to better understand the results.

“The annual rates of chlamydia tests in 2020 (2,355 [95% CI: 2,337-2,373] per 100,000 patients) and 2021 (2,181 [95% CI: 2,162-2,200] per 100,000 patients) were lower than in 2019 (3,592 [95% CI: 3,572-3,613] per 100,000 patients), while the annual rate of gonorrhea tests in 2020 (2,207 [95% CI: 2,187-2,226] per 100,000 patients) was higher than in 2019 (2,129 [95% CI: 2,110-2,147] per 100,000 patients) (S5 Appendix).”

Reviewer: How is chlamydia and gonorrhea tested. Is it not tested together in a PCR test?

Table 1. Age in years

Reviewer: How do you explain the <10 years old with STI test in the data?

Reviewer: See comment before on why testing rates monthly and test positivity quarterly?

Reviewer: Would it be possible to stratify test positivity also by age, sex, rase and ethnicity or are numbers too small?

Reviewer: Fig 4. I would suggest to change the order of the legend accordingly to the order in the graph. Non-Hispanic White, Hispanic, Non-Hispanic Black/African American.

“Relatedly, the increase in the case incidence as well as the various lockdowns during the pandemic period had led to a decrease in primary care practices remaining open or their overall capacity to manage the same patient volumes as they did during the prepandemic period.[25]”

Reviewer: See comment before. I´d suggest to provide background info on times of lockdown.

“Following the guidance, syphilis and gonorrhea were made a top priority by health departments, resulting in the significant decline in chlamydia tests due to the limited resources and the increase in gonorrhea tests in late 2020.[7]”

Reviewer: Why solely gonorrhea and not also chlamydia and syphilis. See comments before. How is chlamydia and gonorrhea tested, separately tested and not together in a PCR test?

“Another potential explanation for the increase in gonorrhea tests in late 2020 is that the stay-at home order ended in many states after June 2020.[30]”

Reviewer: Same question why gonorrhea only and not the other STI tests?

“Black or African American patients were seen at a higher risk of having STIs and related conditions than other racial/ethnic groups.[32]”

Reviewer: Hispanic patients are not seen as at higher risk?

“Another potential limitation is that tests were identified using procedure codes; therefore, tests that were documented in free-text fields were not captured in this study.”

Reviewer: Can you quantify how many and the impact.

6. PLOS authors have the option to publish the peer review history of their article (what does this mean? ). If published, this will include your full peer review and any attached files.

**Do you want your identity to be public for this peer review?** For information about this choice, including consent withdrawal, please see our Privacy Policy .

Reviewer #1: **Yes: ** John Nelson

Reviewer #2: **Yes: ** Daniel Schmidt

---

## [Author Response · Author response to Decision Letter 1]

22 Jul 2024

Reviewer #1: My concerns include the following:

1. Chlamydia (CT) and gonorrhea (GC) can be (and should be) done with swab specimen or urine, so it's not clear why the GC tests were significantly greater than the CT tests. A provider may only order one of the tests but to have such a big difference makes me question the ordering process.

Response: We thank the reviewer for raising these concerns. While we recognize that there exist differences in the chlamydia and gonorrhea testing rates, the database does not provide insights specifically into practice pattern differences in the ordering of tests. Primary care providers’ data are documented from each practice’s electronic health records but does not provide information regarding the process regarding how tests are ordered. We have documented in the manuscript regarding this data limitation, and have added some language into the limitations section of the Discussion. It is important to recognize that future work may require evaluation of documentation standards regarding testing and/or STI co-testing for chlamydia and gonorrhea alongside the clinical decision making and ordering process.

2. CT and GC can be in the vagina/cervix, penis, throat, and/or rectum. It's possible that one patient had a syphilis test, an HIV test, a throat CT/GC test, a rectal CT/GC test, and a urine CT/GC test at the same visit. It's not clear in the manuscript if this was accounted for (extragenital site vs genital site CT/GC tests). And for the positive rates, would a person with a positive GC result of the throat and the rectum on the same day we counted as 1 positive GC or 2 positive GC tests?

Response: We thank the reviewer for highlighting these. We have revised the “Outcome definitions” section to clarify: “A chlamydia or gonorrhea test could be a test from genital, throat, or rectal areas. A person with positive chlamydia or gonorrhea results of genital, throat, or rectal areas on the same day were counted as one positive chlamydia or gonorrhea test.”

3. Why were syphilis and HIV positivity rates not done, and not mentioned in the abstract results? Syphilis is complicated I understand since a positive/reactive test result doesn't necessarily mean an untreated or "new" infection.

Response: We thank the reviewer for raising these concerns. Unlike chlamydia and gonorrhea, it is hard to define a “new” incidence for syphilis and HIV with our data. A presumptive diagnosis of syphilis requires use of two laboratory serologic tests, so one positive test result of syphilis doesn’t necessarily mean a new infection. As for HIV, since it is a lifetime condition and some patients may have already been with it before entering the PRIME Registry, we are unable to determine a new incident HIV simply based on positive test results. This is a limitation of our data source. Given the complexity and uncertainty of calculating the positivity rates for syphilis and HIV, we didn’t include positivity rates for these two conditions in the manuscript.

4. What were the outcomes of the age groups? It would be informative to know the p-value of age group and testing outcomes. Adolescents and young adults have the highest rates of bacterial STIs but were they more or less likely to be tested during COVID?

Response: We agree with the reviewer that it would be informative to know the p-value of age group and testing outcomes. We’ve added the following findings in the Results.

For all four types of STI testing (chlamydia, gonorrhea, syphilis, and HIV), patients aged 10-39 had higher probabilities (P<0.001, Chi-squared test) of being tested from 2019 to 2021 than other age groups.

During the COVID, patients aged 10-24 (i.e. adolescents and young adults) consistently had higher testing rates (P<0.001, Chi-squared test) than others (chlamydia: 4% vs 1% in 2020 and 4% vs 0.9% in 2021; gonorrhea: 4% vs 1% in 2020 and 4% vs 0.9% in 2021; syphilis: 2% vs 0.8% in 2020 and 2% vs 0.7% in 2021; HIV: 3% vs 1% in both 2020 and 2021).

5. Department of Health STI clinics are a common place for people to go for STI testing since it is a public clinic (and testing and treatment is free if not insured) along with family planning clinics - neither of these types of clinics were mentioned on page 5-6 when listing U.S. STI testing site types.

Response: We agree with the reviewer that STI clinics and family planning clinics are common places for STI testing and care. We have added these to the U.S. STI testing site types in the manuscript: “Many healthcare settings are utilized for STI testing in the U.S., including STI clinics, family planning clinics, hospitals, emergency departments, urgent care centers, and physician offices, and differences in how STIs are managed in these different locations have been noted.”

6. Several typos: Human Immunodeficiency Virus should be human immunodeficiency virus on page 5; on abbreviations, you have STI and STIs - I would just use STIs or remove the "s" on infections in the STI write out; Reference #2 has a broken link; page 8, "sexr" should be "sex".

Response: We thank the reviewer for identifying the typos. We have fixed typos on page 5 and page 8. We have removed “STIs-Sexually transmitted infections” from the list of abbreviations. The link in Reference #2 has been fixed.

7. Gay, bisexual, and other MSM may be the reason more "men" were tested for HIV and syphilis compared to women. However, during this time period, and worse now, congenital syphilis rates were increasing, so women should be being tested along with heterosexual men, but none of this is mentioned. I'm assuming there is no sexual orientation identity demographic data in the data set used, but this should be listed as a limitation.

Response: We agree with the reviewer that lack of sexual orientation is a limitation for our study. We mentioned this as a study limitation: “Incomplete demographic data was also a limitation. Sexual orientation and gender diversity information was not available in our data.”

Reviewer #2: Reviewer comments

Thank you for considering me as a reviewer for the study entitled: “Primary care usage patterns associated with sexually transmitted infections during COVID-19” which I read with great interest. The authors examined trends in STI testing and positivity before and during the COVID-19 pandemic in the primary care setting using retrospective data from the PRIME Registry collected from January 1, 2019-December 31, 2021.

Several questions remain that need further clarification please see below.

Response: Thank you for recognizing the contribution of this manuscript. We have revised the manuscript throughout to be more concise and precise, and detail these revisions below.

Abstract

Methods

“We calculated age-standardized monthly and annual STI testing rates and stratified by gender and race and ethnicity.”

Reviewer: I think it would help to clarify which STI you examined.

Response: We thank the reviewer for highlighting this. We’ve added STI categories into the sentence.

“We calculated age-standardized monthly and annual STI testing rates and stratified by sex and race and ethnicity. We also generated quarterly and annual rates for test positivity.”

Reviewer: What was the rational to calculate monthly and annual STI testing rates but for test positivity quarterly and annual rates?

Response: We thank the reviewer for raising these concerns. It was due to the small number of tests that were qualified for the test positivity calculation in our analysis. The small number could restrict us from getting a sense of trend, so we only calculate quarterly and annual rates for test positivity.

Introduction

Reviewer: I think it would be good for the understanding of the results to include information on times of lockdowns in the US.

Response: We added to the Introduction: “Stay-at-home orders were issued in March or April 2020 in the U.S.”

Methods

Reviewer: Based on what did you identify positive test results, how are they coded in the AFC EHR? How likely is miscoding? In S2 Appendix you provided the codes to identify tests, I think it would be good to provide the codes used to identify positive tests. Using for example ICD-10 codes for HIV can be tricky due to miscoding.

Response: AFC EHR has a separate table that includes the results of observations generated by laboratories, imaging procedures, and other procedures. Laboratory results are typically generated by laboratories providing analytic services. These observations are based on analysis of specimens obtained from the patients and submitted to the laboratory. In this table, a typical record includes a patient id, an observation code (in forms of CPT, LOINC, or other lab test codes) and descriptions received from the practice, and the date and value of the result for this observation.

For STI tests, most of the result values were coded as “negative”, “not detected”, “detected”, or “positive”. We determined positive tests as those with result values coded as “detected” or “positive”. We added the following text to “Outcome definition”:

“The evidence of having a positive test was those having results coded as “positive” or “detected” in the EHR.”

We thank the reviewer for highlighting the concern about the miscoding in our data source. Data integrity and coding accuracy in real-world EHR systems could be essential parts that impact patient care. Literature indicated that EHR-related errors could be caused by poor design of the platform, incomplete documentation, and insufficient training on coding (PMID: 38293266; 24159271; 35646364). However, due to the unavailability of the original patients’ files and detailed laboratory test documentation in AFC EHR, we were unable to quantify the miscoding and its potential impact on our analysis. This is a limitation of our study.

Reviewer: Did you distinguish between different stages of Syphilis and latent and acute Syphilis?

Response: We didn’t collect ICD diagnosis codes for syphilis in this study, so we were unable to determine the stage of syphilis or distinguish different stages.

“Patients with missing information for age were excluded.”

Reviewer: I assume this was because you used age for the standardization?

Response: That’s correct. Age was the critical factor to implement the standardization process in our study, so we excluded patients with missing age information.

“Our primary outcome was the monthly and annual rates for testing for chlamydia, gonorrhea, syphilis, and HIV during the study period, …”

“Our secondary outcome was the test positivity for chlamydia and gonorrhea, respectively.”

Reviewer: Why did you only look at chlamydia and gonorrhea for the secondary outcome?

Response: We thank the reviewer for highlighting this. Unlike chlamydia and gonorrhea, it is harder to define a “new” incidence for syphilis and HIV using our data. A presumptive diagnosis of syphilis requires use of two laboratory serologic tests, so one positive test result of syphilis doesn’t necessarily mean a new infection. As for HIV, since it is a lifetime condition and some patients may have already been with it before entering the PRIME Registry, we are unable to determine a new incident HIV simply based on positive test results. This is a limitation of our data source. Given the complexity and uncertainty of calculating the positivity rates for syphilis and HIV, we didn’t include positivity rates for these two conditions in the manuscript.

“Monthly and annual rates of STI tests were calculated and presented for all patients and by category (sex: female and male; race and ethnicity: non-Hispanic Whites, non-Hispanic Blacks or African Americans, Hispanic/Latino).”

Reviewer: Does the data only provide female or male sex, what about sex divers?

Response: Nearly 99% of the patients were labeled as female, male, or unknown in our data. It doesn’t contain further information about sex diversity. We mentioned this as a study limitation: “Incomplete demographic data was also a limitation. Sexual orientation and gender diversity information was not available in our data.”

Results

Reviewer: To avoid confusion I would reorder the sentences and already include the total number of tests in the first paragraph as follows:

A total of 753 primary care practices (S4 Appendix) and 4,410,609 patients were included in the study. Patients had a total of 233,510 tests for chlamydia, 135,559 tests for gonorrhea, 109,909 tests for syphilis, and 135,953 tests for HIV in 2019 – 2021. Among these patients, 180,558 had one or more STI tests (chlamydia: 132,076 patients; 702 practices; gonorrhea: 86,672 patients; 438 practices; syphilis: 69,336 patients; 475 practices; HIV: 90,083 patients; 552 practices) in 2019 – 2021.

Response: We thank the reviewer for the suggestion. We’ve reordered the sentences accordingly.

Trends of STI tests

Reviewer: I would think about including the information into a table as there are a lot of numbers and CI presented and a table would help to compare the different rates over time.

Response: We agree with the reviewer that including a table of numbers and CIs of testing trends would be helpful. We’ve created a table (S3 Table) to show the monthly number of STI tests from 2019 through 2021.

“Rates of all four STI tests further decreased since August 2021 (Fig1).”

Reviewer: See comment on lockdown times to better understand the results.

Response: We agree with the reviewer that adding lockdown times would help to better understand the results. We've included several references (ref [30]-[34]) to the Discussion that illustrate the announcement or extension of the state of emergency due to new waves of COVID-19 cases in the U.S. during 2021. These could help to explain the reduction in STI testing rates that we observed from our data since August 2021.

“The annual rates of chlamydia tests in 2020 (2,355 [95% CI: 2,337-2,373] per 100,000 patients) and 2021 (2,181 [95% CI: 2,162-2,200] per 100,000 patients) were lower than in 2019 (3,592 [95% CI: 3,572-3,613] per 100,000 patients), while the annual rate of gonorrhea tests in 2020 (2,207 [95% CI: 2,187-2,226] per 100,000 patients) was higher than in 2019 (2,129 [95% CI: 2,110-2,147] per 100,000 patients) (S5 Appendix).”

Reviewer: How is chlamydia and gonorrhea tested. Is it not tested together in a PCR test?

Response: We thank the reviewer for raising these concerns. Our results show that the overall testing rate for chlamydia was higher than gonorrhea in 2019, and the two testing rates were getting closer in 2020 and 2021. While we recognize that there exist differences in the chlamydia and gonorrhea testing rates, the database does not provide insights specifically into practice pattern differences in the ordering of tests. Primary care providers’ data are documented from each practice’s electronic health records but does not provide information regarding the process regarding how tests are ordered. We have added some language into the limitations section of the Discussion.

Table 1. Age in years

Reviewer: How do you explain the <10 years old with STI test in the data?

Response: In our data, around 0.2% of the patients were less than 10 when they had STI tests. Reasons for testing in this age group included child sexual abuse or suspected exposure to sexually transmitted infections. Another possible reason could be data entry error, such as clinicians providing incorrect dates of birth.

Reviewer: See comment before on why testing rates monthly and test positivity quarterly?

Response: We thank the reviewer for raising these concerns. It was due to the small number of tests that were qualified for the test positivity calculation in our analysis. The small number could restrict us from getting a sense of trend, so we only calculate quarterly and annual rates for test positivity.

Reviewer: Would it be possible to stratify test positivity also by age, sex, race and ethnicity or are numbers too small?

Response: The numbers of tests for each STI qualified for test positivity calculation would be small after stratification, which might prevent meaningful statistical analysis.

Reviewer: Fig 4. I would suggest to change the order of the legend accordingly to the order in the graph. Non-Hispanic White, Hispanic, Non-Hispanic Black/Af

---

## [Decision Letter · Decision Letter 1]

20 Feb 2025

PONE-D-24-07258R1Primary care usage patterns associated with sexually transmitted infections during COVID-19PLOS ONE

Dear Dr. Hao,

Thank you for submitting your manuscript to PLOS ONE. After careful consideration, we feel that it has merit but does not fully meet PLOS ONE’s publication criteria as it currently stands. Therefore, we invite you to submit a revised version of the manuscript that addresses the points raised during the review process.

The reviewers offer highly informed suggestions and I agree with all of them. Please address them, remembering that there is an international readership that may differ in policies and practices.

We look forward to receiving your revised manuscript.

Kind regards,

Janet E Rosenbaum, Ph.D.

Academic Editor

PLOS ONE

Journal Requirements:

Reviewers' comments:

Reviewer's Responses to Questions

**Comments to the Author**

1. If the authors have adequately addressed your comments raised in a previous round of review and you feel that this manuscript is now acceptable for publication, you may indicate that here to bypass the “Comments to the Author” section, enter your conflict of interest statement in the “Confidential to Editor” section, and submit your "Accept" recommendation.

Reviewer #3: (No Response)

Reviewer #4: (No Response)

2. Is the manuscript technically sound, and do the data support the conclusions?

Reviewer #3: Yes

Reviewer #4: Yes

3. Has the statistical analysis been performed appropriately and rigorously? 

Reviewer #3: Yes

Reviewer #4: Yes

4. Have the authors made all data underlying the findings in their manuscript fully available?

Reviewer #3: No

Reviewer #4: Yes

5. Is the manuscript presented in an intelligible fashion and written in standard English?

Reviewer #3: Yes

Reviewer #4: Yes

6. Review Comments to the Author

Reviewer #3: This article presents an interesting study on the changes in monthly and annual testing rates for chlamydia, gonorrhea, syphilis, and HIV in primary care practices in the US during the COVId-19 pandemic compared to pre-pandemic levels.

The study shows a significant drop in STIs testing in the primary healthcare setting, particularly during stay-at-home orders, followed by a recovery after the relaxation of those orders, although testing rates did not return to 2019 levels. Interestingly, gonorrhea testing rates increased in 2020 compared to 2019, and females and non-Hispanic Black or African Americans patients exhibited higher annual testing rates.

The manuscript is generally well written and clearly and logically presented. However, in my view, there are some areas that require review and revision before it can be accepted for publication. Please find my comments below:

• Abstract

o The authors should specify in the abstract the country/region where the study was conducted.

o The results section of the abstract should include some numerical data, rather that only stating increases or decreases.

o In the last sentence of the results section, “Test positivity during the pandemic was elevated for both chlamydia and gonorrhea”, the term ‘elevated’ is unclear. what does mean “elevated”? he authors should provide specific numbers and clarify the comparison baseline.

• Introduction

(No specific comments)

• Methods

o The secondary outcome was the test positivity for chlamydia and gonorrhea. Why weren’t test positivity rates for syphilis and HIV included as secondary outcomes?

o In the first line of the Patient Characteristics section, eliminate the extra ‘r’ in ‘sexr’.

• Results

(No specific comments)

• Discussion

o Overall, I found the discussion section too brief. It would benefit from, a comparison of these results with studies conducted in other setting or in other countries/regions. This would enable a deeper understanding of discussion of the hypotheses regarding the increase in gonorrhea testing rates compared to other STIs, as well as the higher testing rates among females and non-Hispanic Black or African American patients.

o The stated limitation about symptom presentation at the time of testing, which prevents distinguish between tests conducted for screening versus diagnosis, is critical. While this is an important limitation, the discussion could be enriched by incorporating findings from other studies. For instance, the authors could address how the higher symptom presentation for gonorrhea compared to chlamydia, as well as the potential role of pregnancy screening (as mentioned by the authors), might have influenced the results.

• Conclusion

o The authors should mention the country/region where the study has been conducted.

Reviewer #4: Thank you for the opportunity to review this paper, that has been previously reviewed and commented on. I have checked the previous reviewers comments, which the authors seem to have responded to appropriately. The only outstanding concern I have which is shared by the other two reviewers. In New Zealand the PCR test we use to test for Chlamydia and Gonorrhea tests for both infections at the same time. The health provider is not able to chose only one. So you will never recieve test results for Chlamydia only or Gonorrhea only, it will always contain two results. I'm not sure if its the same in the US, but your previous reviewers are questioning why you would have more results in your database for Gonoorhea than you would for Chlamydia when both are tested for at the same time (and both results reported). I don't think you have answered this satisfactorily. Check with the lab whether this is the case and if it is you will need to try and find out why you have more results for one test than the other. If the lab tests them separately then make this explicit so that overseas readers like me are not left wondering about the descrepancy.

Some further suggestions to aid with clarity of the manuscript:

1. when you talk about race and ethnicity I think it would be better to group them together as "race/ethnicity". At the moment it sounds like you have two separate measures, one for race and one for ethnicity when I think you only have one data point, is that correct?

2. Table one - you could remove most of the repeated headings for each column and put them in the title instead. So take out: "patients who had" and "2019-2021" and change the table heading to something like: Characteristics of patients who had STI tests, American Family Cohort, 2019-2021. The column with no testing could be "No record of STI testing".

3. Could you make it explicit in the title of the paper, or the abstract that the study was carried out in the US?

4. I'm not sure whether I agree with Primary Care as a locus of monitoring mentioned in your conclusion. Its focus is really to screen, diagnose and treat. Its more that primary care data provides an opportunity for monitoring.

7. PLOS authors have the option to publish the peer review history of their article (what does this mean? ). If published, this will include your full peer review and any attached files.

**Do you want your identity to be public for this peer review?** For information about this choice, including consent withdrawal, please see our Privacy Policy .

Reviewer #3: No

Reviewer #4: No

---

## [Author Response · Author response to Decision Letter 2]

11 Mar 2025

Journal Requirements:

Response: We’ve reviewed and updated the references in the revised manuscript. Specifically, we replaced the original references 2, 18-20, 27, and 29 with new ones as their web links are no longer valid or might be invalid soon. All references in the revised manuscript are now complete and correct.

Comments to the Author

Reviewer #3: This article presents an interesting study on the changes in monthly and annual testing rates for chlamydia, gonorrhea, syphilis, and HIV in primary care practices in the US during the COVId-19 pandemic compared to pre-pandemic levels.

The study shows a significant drop in STIs testing in the primary healthcare setting, particularly during stay-at-home orders, followed by a recovery after the relaxation of those orders, although testing rates did not return to 2019 levels. Interestingly, gonorrhea testing rates increased in 2020 compared to 2019, and females and non-Hispanic Black or African Americans patients exhibited higher annual testing rates.

The manuscript is generally well written and clearly and logically presented. However, in my view, there are some areas that require review and revision before it can be accepted for publication. Please find my comments below:

Abstract

The authors should specify in the abstract the country/region where the study was conducted.

Response: We added this information to the Abstract: “This study examined trends in STI testing and positivity before and during the COVID-19 pandemic in the primary care setting in the United States.”

The results section of the abstract should include some numerical data, rather that only stating increases or decreases.

In the last sentence of the results section, “Test positivity during the pandemic was elevated for both chlamydia and gonorrhea”, the term ‘elevated’ is unclear. what does mean “elevated”? The authors should provide specific numbers and clarify the comparison baseline.

Response: We thank the reviewer for highlighting this. We modified the result section as:

“We observed a substantial decline in testing rates for STIs from January-April 2020 (53% for chlamydia, 46% for gonorrhea, 36% for syphilis, 42% for HIV), followed by a rapid increase in May-June 2020 (64% for chlamydia, 65% for gonorrhea, 32% for syphilis, 48% for HIV). Chlamydia testing rates decreased from 2019 to 2020 (2,355 vs 3,592, per 100,000), while gonorrhea testing rates increased during this time (2,207 vs 2,129, per 100,000). STI testing rates for females and non-Hispanic Black or African American patients were higher than for other groups during the pandemic. Test positivity was higher in Q2-Q4 in 2020 than in 2019 for both chlamydia (1.6% vs 1.4%) and gonorrhea (0.5% vs 0.3%).”

Introduction

(No specific comments)

Methods

The secondary outcome was the test positivity for chlamydia and gonorrhea. Why weren’t test positivity rates for syphilis and HIV included as secondary outcomes?

Response: We thank the reviewer for raising this concern.

Unlike chlamydia and gonorrhea, it is hard to confirm a “new” incidence for syphilis and HIV with our data. A presumptive diagnosis of syphilis requires use of two laboratory serologic tests, so one positive test result of syphilis doesn’t necessarily mean a new infection. For HIV, since it is a lifetime condition and some patients may have been diagnosed outside the PRIME Registry network, we are unable to determine a new incident HIV simply based on positive test results documented in our data. These issues are related to data integrity and interoperability, underscoring challenges of EHR-based studies. Given the complexity and uncertainty of calculating the positivity rates for syphilis and HIV, we didn’t include positivity rates for these two conditions in the manuscript.

In the first line of the Patient Characteristics section, eliminate the extra ‘r’ in ‘sexr’.

Response: We’ve changed it to “gender”.

Results

(No specific comments)

Discussion

Overall, I found the discussion section too brief. It would benefit from, a comparison of these results with studies conducted in other setting or in other countries/regions. This would enable a deeper understanding of discussion of the hypotheses regarding the increase in gonorrhea testing rates compared to other STIs, as well as the higher testing rates among females and non-Hispanic Black or African American patients.

Response: We thank the reviewer for highlighting this. We revised the third paragraph in Discuss by adding a comparison with other studies:

“Notably, some US-based studies reported a decrease in gonorrhea testing volumes in 2020 compared to 2019 [33, 34]. This might reflect variations in resource allocation patterns across different clinical settings during the pandemic. This might also indicate an improvement in gonorrhea test documentation within structured EHR systems in PRIME practices, which could result in higher observed gonorrhea testing rates.”

We also revised the 4th paragraph in Discussion:

“Our findings that females and non-Hispanic Black or African American patients having higher annual STI testing rates during our study period are consistent with the results of a few studies.[33-37] This may be explained by several phenomena. The high rate amongst females could be due to the U.S. Preventive Services Task Force (USPSTF) recommendations for STI screening for sexually active young women or due to pregnancy screening.[38] However, there is insufficient evidence in our study to suggest if STI testing is screening or diagnosis. Studies based on EHR, surveys, and claims data also found that Black or African American patients were more likely to receive STI tests compared to Whites.[35-37, 39] This is potentially related to the finding that this population were seen at a higher risk of having STIs and related conditions than White patients,[2, 40] which might have led to a risk-based prioritization in providing STI testing services.[39] Further, research within a U.S. hospital system found that despite a reduction in overall STI testing counts during the early pandemic, females and non-Hispanic Black or African American patients had increased testing proportions compared to males and Whites.[33] These findings, together with ours, may reflect that the clinical settings prioritized certain populations for STI testing in response to the public health emergency.[5]”

The stated limitation about symptom presentation at the time of testing, which prevents distinguishing between tests conducted for screening versus diagnosis, is critical. While this is an important limitation, the discussion could be enriched by incorporating findings from other studies. For instance, the authors could address how the higher symptom presentation for gonorrhea compared to chlamydia, as well as the potential role of pregnancy screening (as mentioned by the authors), might have influenced the results.

Response: We added the following sentences to the study limitation to discuss the role of symptom presentation and pregnancy screening:

“Furthermore, we do not know symptom presentation at the time of testing from the procedure codes; therefore, we are unable to discern tests for screening versus diagnosis or investigate how symptom presentation influenced the relationship between patient-level characteristics and testing rates. Additionally, given that a large proportion of chlamydia and gonorrhea cases are asymptomatic,[41-43] routine screening for STIs is essential for early identification and treatment.[41, 43] We are unable to examine the proportion of routine screening, though. We are also unable to access whether clinics prioritized tests to symptomatic patients due to limited resources during the COVID.[33, 34]”

“Pregnancy-related data were not collected in this study. The role of pregnancy-related screening in testing rates and the adherence to USPSTF and CDC’s recommendation,[38, 44, 45] therefore, is unknown.”

Conclusion

The authors should mention the country/region where the study has been conducted.

Response: We modified the conclusion to highlight the place where the study was conducted: “This study investigated the monthly and annual testing rates for chlamydia, gonorrhea, syphilis, and HIV in the U.S. primary care practices.” We also modified the title and abstract accordingly.

Reviewer #4: Thank you for the opportunity to review this paper, that has been previously reviewed and commented on. I have checked the previous reviewers’ comments, which the authors seem to have responded to appropriately. The only outstanding concern I have which is shared by the other two reviewers. In New Zealand the PCR test we use to test for Chlamydia and Gonorrhea tests for both infections at the same time. The health provider is not able to choose only one. So you will never receive test results for Chlamydia only or Gonorrhea only, it will always contain two results. I'm not sure if it’s the same in the US, but your previous reviewers are questioning why you would have more results in your database for Gonorrhea than you would for Chlamydia when both are tested for at the same time (and both results reported). I don't think you have answered this satisfactorily. Check with the lab whether this is the case and if it is you will need to try and find out why you have more results for one test than the other. If the lab tests them separately then make this explicit so that overseas readers like me are not left wondering about the discrepancy.

Response: Thank you for bringing this to our attention.

While chlamydia and gonorrhea are often tested together in primary care in the United States, the data source used for this study documented these tests separately. In this study, over 60% of the chlamydia tests were documented using chlamydia-specific CPT codes 87490 and 87491, and over 85% of the gonorrhea tests were documented using gonorrhea-specific CPT codes 87590 and 87591. Additionally, 53% of the chlamydia tests and 92% of the gonorrhea tests can be matched together based on the test dates and patient information, suggesting that these tests were ordered together for the same patients. We were unable to map the rest 47% of the chlamydia tests to any gonorrhea tests, however. Further, in our data, 35% of the practices documented chlamydia tests but didn’t record any gonorrhea tests during the same period. These resulted in a large difference in counts between the two tests, as shown in Table 1.

The discrepancy in counts could be likely related to practice-level factors. These practices might not have submitted gonorrhea test files to our database, or the files they submitted were filtered out by the system due to issues like data quality. Additionally, gonorrhea-related information might be recorded in unstructured formats or in notes, which are currently not accessible to us.

We’ve added a footnote to Table 1 to outline these potential reasons: “Chlamydia and gonorrhea tests are often ordered together in primary care. However, in our data, a few practices documented chlamydia tests in the EHR but didn’t document any gonorrhea tests throughout the entire study period, resulting in a difference in total counts between the two tests. The absence of gonorrhea test documents from these practices could be due to variations in EHR data quality and integrity, which limits our access to them.”

Some further suggestions to aid with clarity of the manuscript:

1. when you talk about race and ethnicity I think it would be better to group them together as "race/ethnicity". At the moment it sounds like you have two separate measures, one for race and one for ethnicity when I think you only have one data point, is that correct?

Response: Thanks for pointing this out. We’ve changed “race and ethnicity” to “race/ethnicity” throughout the manuscript.

2. Table one - you could remove most of the repeated headings for each column and put them in the title instead. So take out: "patients who had" and "2019-2021" and change the table heading to something like: Characteristics of patients who had STI tests, American Family Cohort, 2019-2021. The column with no testing could be "No record of STI testing".

Response: Thank you. We’ve updated Table 1 accordingly.

3. Could you make it explicit in the title of the paper, or the abstract that the study was carried out in the US?

Response: To highlight the country where the study was carried out, we changed the title to: “Primary care usage patterns associated with sexually transmitted infections in the United States during COVID-19.” We also added this to the Abstract: “This study examined trends in STI testing and positivity before and during the COVID-19 pandemic in the primary care setting in the United States.”

4. I'm not sure whether I agree with Primary Care as a locus of monitoring mentioned in your conclusion. Its focus is really to screen, diagnose and treat. Its more that primary care data provides an opportunity for monitoring.

Response: We agree with the reviewer’s point and have modified the abstract and conclusion accordingly: “This study emphasizes the role of primary care EHR data in monitoring and an opportunity for closer collaboration with public health agencies.”

---

## [Editor Report · Decision Letter 2]

14 Mar 2025

PONE-D-24-07258R2Primary care usage patterns associated with sexually transmitted infections in the United States during COVID-19PLOS ONE

Dear Dr. Hao,

Thank you for submitting your manuscript to PLOS ONE. After careful consideration, we feel that it has merit but does not fully meet PLOS ONE’s publication criteria as it currently stands. Therefore, we invite you to submit a revised version of the manuscript that addresses the points raised during the review process.

You have addressed reviewers' suggestions. I have had the chance to make a comprehensive review that complements the reviewers' suggestions with regard to title, abstract, analysis, and results. As you change these elements, you may also want to revisit the other portions of the paper including the discussion.

Suggestions for the title and abstract:

1. The short title of the paper is clearer than the full title.

Full Title: Primary care usage patterns associated with sexually transmitted infections in the United States during COVID-19

Short Title: STI screening in U.S. primary care during COVID-19

Please rewrite the full title to more clearly reflect the research question’s exposure variable. The full title suggests that the primary exposure variable is primary care usage patterns, such as number of visits per year, such as comparing patients who made 2+ visits per year with 1 visit per year. It seems from the results that your exposure variable is the year 2020 versus the year 2019. With the three years 2019-21, it seems that you have three periods: before covid-19, covid-19 before the vaccine, and covid-19 after vaccine. Either use 2 periods (before and after covid) or 3 periods (before, pre-vaccine, post-vaccine).

2. Sample sizes go in the methods section: n=180558 patients tested for STIs out of 4410609 patients seen at 753 primary care practices over 3 years.

3. You have 3 years of data: 2019, 2020, and 2021, but the entire results section of the abstract only refers to the first two years. Please write the abstract so that the methods section says specifically 3 years and so that in results section you are referring more clearly to the data. I don’t see any references at all to any data from January 1 2021 to December 31 2021. I might expect each result to have 3 numbers, one for each of the years of the data. Please also refer to the years in the title of the paper.

4. The title refers to “during covid-19” and in the results section of the abstract you refer to three different time increments: months, quarters, and years. Statistically, using multiple time periods is a concern because making multiple comparisons can cause false significance.

More specific to the subject matter, the time divisions do not correspond naturally to the subject matter of the covid-19 pandemic. For example, you note that there is a decrease in January to April 2020 followed by an increase in May-June, but the rationale for this time division is not clear because January and February 2020 and the first week or two of March are not “during COVID-19” in any meaningful sense. We would not expect that time period to have different STI testing patterns. Please use a time division that corresponds to start of pandemic and optionally a second division after vaccine availability.

5. Please give a couple of words describing the PRIME registry for readers who have never heard of it before.   

6. Maintain parallelism. In the abstract results section, you give the years 2019 and 2020 in that order, but in parentheses you list the number of people tested for chlamydia and gonorrhea in 2020 followed by 2019. As noted above, though, I think you should have well-defined and meaningful time periods with respect to covid-19.

7. The methods section must specify any statistical tests that you used, and I don’t see any statistical analysis methods referred to. You calculated the age-adjusted testing rates, but you do not refer to an analysis method to evaluate whether one rate differs from another.

Analysis:

1. Age adjustment: please be specific about which US Census data was used for age adjustment. I see from figure legends that you adjusted by US population. Please specify US country-level age adjustment or a less clunky term to be clear. Otherwise I would have thought each clinic had a catchment area and you adjusted by the county age distribution for each location or similar.

2. Keyfitz formula is age-adjusted rate divided by square root of the number of events. It is not complicated, please actually state that instead of making the reader look it up. You do not show the error bars on your figures, so it is hard to tell what is different.

It may be more natural to at least attempt a basic time series analysis. Perhaps you will not see anything, but sometimes you can see a pattern in these pre vs during covid data. (We did this in our publications on pediatric GI diseases.) I see a decrease in both Januaries without covid in 2019 and 2020, and maybe there is a seasonal component to STI testing.

3. Figures are nice, but I don’t think either figure 1 or 2 is the optimal data display. Consider modifying figure 1 so that it separates the diseases, so make figure 1 into 4 plots. That will give you space to put the error bars on each point or some other means of communicating to the reader what is different.  I suggest further putting dotted line markers at time points that are meaningful for covid pandemic. If you want 2 time periods put one somewhere in March 2020 and if you want 3, put a second line at a meaningful time with respect to vaccine availability such as April 2021.

4. You have an enormous amount of supplementary information. Given PLOS ONE has pretty minimal manuscript requirements with no word limits or figure limits, much of the information can be incorporated into the paper and then the supplementary information will be minimized or eliminated. Readers do not in practice read as much of the supplementary information as much as they read the paper. When faculty distribute published papers to their students, the supplementary information is not included.

     a. Information in S2 file seems pertinent to your methods section. Please include more of that information in paper body.

    b. S2 table is better as a map. R will produce a US map with numbers in the states. Here are some options: https://jtr13.github.io/cc19/different-ways-of-plotting-u-s-map-in-r.html

I think other software tools will do this as well.

    c. S3 table would be good to include as a table within the paper itself as opposed to supplementary.  Another way to arrange the data that may be clearer is by disease rather than by year, so that the columns are 2019, 2020, and 2021, allowing comparison of chlamydia across all the March and April, for example.

    d. Figure 4 and S1 and S2 figures overlap a lot. I suggest just making figure 4 and 5, where figure 4 is all diseases by gender and figure 5 is all diseases by race/ethnicity so you can take out the corresponding supplementary figures.

We look forward to receiving your revised manuscript.

Kind regards,

Janet E Rosenbaum, Ph.D.

Academic Editor

PLOS ONE
---

## [Author Response · Author response to Decision Letter 3]

11 Apr 2025

Suggestions for the title and abstract:

1. The short title of the paper is clearer than the full title.

Full Title: Primary care usage patterns associated with sexually transmitted infections in the United States during COVID-19

Short Title: STI screening in U.S. primary care during COVID-19

Please rewrite the full title to more clearly reflect the research question’s exposure variable. The full title suggests that the primary exposure variable is primary care usage patterns, such as number of visits per year, such as comparing patients who made 2+ visits per year with 1 visit per year. It seems from the results that your exposure variable is the year 2020 versus the year 2019. With the three years 2019-21, it seems that you have three periods: before covid-19, covid-19 before the vaccine, and covid-19 after vaccine. Either use 2 periods (before and after covid) or 3 periods (before, pre-vaccine, post-vaccine).

Response: We thank the Editor for raising this. Our analysis covers a three-year period from 2019 to 2021 in the US. Since this period was across the onset of COVID-19 and the stay-at-home order in March 2020, we would like to investigate how COVID-19 impacts the annual and monthly patterns of STI testing rates, plus the test positivity. While we are cognizant that there are three distinct periods, our analysis primarily wishes to explore these trends across the entire time frame with some attention to each of the three periods (before COVID, pre-vaccine during COVID, and post-vaccine during COVID).

We’ve updated the full title as “Primary care screening for sexually transmitted infections in the United States from 2019 to 2021”.

2. Sample sizes go in the methods section: n=180558 patients tested for STIs out of 4410609 patients seen at 753 primary care practices over 3 years.

Response: We moved the sample size description to the Methods: “753 primary care practices and 4,410,609 patients were included, with 180,558 having one or more STI tests.”

3. You have 3 years of data: 2019, 2020, and 2021, but the entire results section of the abstract only refers to the first two years. Please write the abstract so that the methods section says specifically 3 years and so that in results section you are referring more clearly to the data. I don’t see any references at all to any data from January 1 2021 to December 31 2021. I might expect each result to have 3 numbers, one for each of the years of the data. Please also refer to the years in the title of the paper.

Response: We thank the Editor for highlighting this.

We rewrote the Results section by adding data in 2021: “Testing rates per 100,000 decreased from 2019 to 2021 for chlamydia (3,592 vs 2,355 vs 2,181) while increased for gonorrhea in 2020 (2,129 vs 2,207 vs 2,057). STI testing rates from 2019 to 2021 for females and non-Hispanic Black or African American patients were higher than other groups.”

4. The title refers to “during covid-19” and in the results section of the abstract you refer to three different time increments: months, quarters, and years. Statistically, using multiple time periods is a concern because making multiple comparisons can cause false significance.

More specific to the subject matter, the time divisions do not correspond naturally to the subject matter of the covid-19 pandemic. For example, you note that there is a decrease in January to April 2020 followed by an increase in May-June, but the rationale for this time division is not clear because January and February 2020 and the first week or two of March are not “during COVID-19” in any meaningful sense. We would not expect that time period to have different STI testing patterns. Please use a time division that corresponds to start of pandemic and optionally a second division after vaccine availability.

Response: We revised the time division for STI testing rates at early pandemic: “We observed a substantial decline in testing rates for STIs from March-April 2020 (31% for chlamydia, 30% for gonorrhea, 23% for syphilis, 24% for HIV)”. We also rewrote the test positivity results in annual trend for consistency: “An increase in test positivity from 2019 to 2021 was observed for gonorrhea (0.4% vs 0.4% vs 0.5%) but no significant change for chlamydia (1.5% vs 1.6% vs 1.5%).”

We also updated the title and Background section accordingly to highlight the study period: “This study examined trends in STI testing and positivity from 2019 to 2021 in primary care in the United States.”

5. Please give a couple of words describing the PRIME registry for readers who have never heard of it before.

Response: We revised the first sentence in Methods: “This is a retrospective study using the PRIME Registry, a national primary care EHR registry, from January 1, 2019-December 31, 2021.”

6. Maintain parallelism. In the abstract results section, you give the years 2019 and 2020 in that order, but in parentheses you list the number of people tested for chlamydia and gonorrhea in 2020 followed by 2019. As noted above, though, I think you should have well-defined and meaningful time periods with respect to covid-19.

Response: We thank the Editor for highlighting this. We’ve revised this and presented the testing rates in an order of 2019, 2020, 2021 for chlamydia and gonorrhea. We also included the drop in % for all four STIs from March to April 2020.

7. The methods section must specify any statistical tests that you used, and I don’t see any statistical analysis methods referred to. You calculated the age-adjusted testing rates, but you do not refer to an analysis method to evaluate whether one rate differs from another.

Response: We added at the end of Methods: “Chi-square tests and 95% confidence intervals were used for comparison.”

Analysis:

1. Age adjustment: please be specific about which US Census data was used for age adjustment. I see from figure legends that you adjusted by US population. Please specify US country-level age adjustment or a less clunky term to be clear. Otherwise I would have thought each clinic had a catchment area and you adjusted by the county age distribution for each location or similar.

Response: We’ve added the source data we used for age adjustment: “Direct age adjustment was based on using the reference U.S. age distribution per the U.S. Census 2020 data for the study period. This is based on Annual Estimates of the Resident Population by Sex, Age, Race, and Hispanic Origin for the United States: April 1, 2020 to July 1, 2021.”

2. Keyfitz formula is age-adjusted rate divided by square root of the number of events. It is not complicated, please actually state that instead of making the reader look it up. You do not show the error bars on your figures, so it is hard to tell what is different.

It may be more natural to at least attempt a basic time series analysis. Perhaps you will not see anything, but sometimes you can see a pattern in these pre vs during covid data. (We did this in our publications on pediatric GI diseases.) I see a decrease in both Januaries without covid in 2019 and 2020, and maybe there is a seasonal component to STI testing.

Response: We added the formula for calculating 95% confidence intervals of age-adjusted rates:

“The 95% confidence interval (CI) was calculated based on the standard errors generated using the Keyfitz formula:

95% CI= ±1.96×R/√N

where R is the adjusted rate for STI testing, N is the number of tests.”

We agree with the Editor that time series analysis would allow us to further explore seasonal patterns of STI tests in primary care, providing insights into underlying potential drivers. Testing behaviors can be influenced by multiple factors. Some patients may come for tests due to symptoms and seek an STI diagnosis, while others may be asymptomatic but at increased risk. Certain groups of patients may also come for annual routine screening, like women aged 15 to 24 who are sexually active. Seasonal patterns of tests, therefore, can be driven by various motivations. However, due to the limitation in our data source, we are unable to differentiate a routine screening from a diagnostic test or identify the motivations for screening. Further expounding upon potential limitations to draw inference about this variation is based on not knowing lifestyle and risky sexual behaviors as being drivers of this variation across time since these are not captured in the PRIME Registry in any reliable form. Lifestyle and sexual behaviors (or risk taking) could influence screening motivation or the patient’s suspicion that they may have an STI thereby driving up screening in this population. Due to these limitations, it is tenuous to link seasonal patterns of tests to seasonal patterns of infections. We plan to include this analysis in future studies using external linkage datasets with increasing patient details which can help explain these variations in testing and test positivity.

3. Figures are nice, but I don’t think either figure 1 or 2 is the optimal data display. Consider modifying figure 1 so that it separates the diseases, so make figure 1 into 4 plots. That will give you space to put the error bars on each point or some other means of communicating to the reader what is different. I suggest further putting dotted line markers at time points that are meaningful for covid pandemic. If you want 2 time periods put one somewhere in March 2020 and if you want 3, put a second line at a meaningful time with respect to vaccine availability such as April 2021.

Response: We’ve revised Fig 1 accordingly. Testing rates for each STI were plotted separately with error bars. A dotted line was added to indicate the beginning of stay-at-home order in the U.S. in March 2020.

4. You have an enormous amount of supplementary information. Given PLOS ONE has pretty minimal manuscript requirements with no word limits or figure limits, much of the information can be incorporated into the paper and then the supplementary information will be minimized or eliminated. Readers do not in practice read as much of the supplementary information as much as they read the paper. When faculty distribute published papers to their students, the supplementary information is not included.

a. Information in S2 file seems pertinent to your methods section. Please include more of that information in paper body.

Response: S1 file and S2 file were incorporated into the main text.

b. S2 table is better as a map. R will produce a US map with numbers in the states. Here are some options: https://jtr13.github.io/cc19/different-ways-of-plotting-u-s-map-in-r.html

I think other software tools will do this as well.

Response: We plotted the data from S2 Table as a map and renamed it as S1 Fig.

c. S3 table would be good to include as a table within the paper itself as opposed to supplementary. Another way to arrange the data that may be clearer is by disease rather than by year, so that the columns are 2019, 2020, and 2021, allowing comparison of chlamydia across all the March and April, for example.

Response: We re-arranged the data in S3 Table by STI and moved it to the main text (Table 2).

d. Figure 4 and S1 and S2 figures overlap a lot. I suggest just making figure 4 and 5, where figure 4 is all diseases by gender and figure 5 is all diseases by race/ethnicity so you can take out the corresponding supplementary figures.

Response: We combined Fig 4 and S1 Fig into a new Fig 4 and Fig 5.

---

## [Editor Report · Decision Letter 3]

6 May 2025

Primary care screening for sexually transmitted infections in the United States from 2019 to 2021

PONE-D-24-07258R3

Dear Dr. Hao,

We’re pleased to inform you that your manuscript has been judged scientifically suitable for publication and will be formally accepted for publication once it meets all outstanding technical requirements.

Kind regards,

Janet E Rosenbaum, Ph.D.

Academic Editor

PLOS ONE
---

## [Editor Report · Acceptance letter]

PONE-D-24-07258R3

PLOS ONE

Dear Dr. Hao,

I'm pleased to inform you that your manuscript has been deemed suitable for publication in PLOS ONE. Congratulations! Your manuscript is now being handed over to our production team.

Kind regards,

on behalf of

Dr. Janet E Rosenbaum

Academic Editor

PLOS ONE